

# The influence of ammonia emissions on the size-resolved global atmospheric aerosol composition and acidity

Xurong Wang[1], Alexandra P. Tsimpidi[1], Zhenqi Luo[2], Benedikt Steil[3], Andrea Pozzer[3], Jos Lelieveld[3,4], and Vlassis A. Karydis[1]

[1] Institute of Climate and Energy Systems: Troposphere (ICE-3), Forschungszentrum Jülich GmbH, Jülich, Germany
[2] School of Integrative Plant Science, Soil and Crop Sciences Section, Cornell University, Ithaca, NY 14853, United States of America
[3] Max Planck Institute for Chemistry, Atmospheric Chemistry Dept., Mainz, Germany.
[4] The Cyprus Institute, Climate and Atmosphere Research Center Nicosia, Nicosia, Cyprus.

*Correspondence to:*
Vlassis A. Karydis (v.karydis@fz-juelich.de) and Alexandra P. Tsimpidi (a.tsimpidi@fz-juelich.de)

**Abstract.** Ammonia ($NH_3$) is an abundant alkaline gas in the atmosphere and a key precursor in the formation of particulate matter. While emissions of other aerosol precursors such as $SO_2$ and $NO_x$ have decreased significantly, global $NH_3$ emissions are stable or increasing, and this trend is projected to continue. This study investigates the impact of $NH_3$ emission changes on size-resolved aerosol composition and acidity using the atmospheric chemistry-climate model EMAC. Three $NH_3$ emission schemes are analyzed: two bottom-up inventories and one derived using an updated top-down method. The results reveal that sulphate-nitrate-ammonium aerosols in two fine mode size ranges (0-1 µm and 1-2.5 µm) show the greatest sensitivity to $NH_3$ emission changes. Regional responses vary depending on the local chemical environment of secondary inorganic aerosols. In 'NH₃-rich' regions (e.g. East Asia and Europe), the abundance of $NH_3$ partially offsets the effects of reduced $NH_3$ emissions when $NO_x$ and $SO_2$ are available, especially for aerosols in the 1-2.5 µm range. This highlights the importance of coordinated control strategies for $NH_3$, $NO_x$ and $SO_2$ emissions. Further, we find that $NH_3$ has a buffering effect in densely populated areas, maintaining aerosol acidity at moderate levels and mitigating drastic pH shifts. The study emphasizes that pH changes are closely related to $NH_3$ emission variations, with the highest sensitivity observed in the fine mode size ranges. These results highlight the critical role of $NH_3$ in shaping aerosol acidity and argue for size-specific approaches to managing particulate matter.

## 1. Introduction

As an abundant alkaline gas in the atmosphere, ammonia ($NH_3$) acts as a precursor in the formation of particulate matter by neutralizing atmospheric acids (e.g., $H_2SO_4$, $HNO_3$) to form sulfate-nitrate-ammonium (SNA) aerosols (Li et al., 2018; Chen et al., 2016; Wang et al., 2013), which are the main secondary inorganic components of $PM_{2.5}$ (particulate matter with a diameter of 2.5 µm or less). By condensing onto freshly nucleated particles, $NH_3$ enhances the growth rate of new particles as well as their hydrophilicity (Wang et al., 2020; Li et al., 2018). This can degrade air quality and change the solar radiative balance by interaction with radiation and clouds (Che et al., 2009; Zhao et al., 2011; Yao et al., 2018). In addition, the conjugate base-acid pair $NH_3/NH_4^+$ acts as the major buffer that inhibits changes in aerosol acidity (Chen et al., 2019; Zheng et al., 2020; Karydis et al., 2021).



Anthropogenic emissions are the main source of atmospheric $NH_3$, with an average global contribution of 76%,

dominated by agricultural activities, including livestock farming and the fertilization of soils (Schlesinger and Hartley, 1992; Dentener and Crutzen, 1994; Bouwman et al., 1997; Olivier et al., 1998; Van Aardenne et al., 2001; Bleeker et al., 2013). Meanwhile, the importance of non-agricultural sources, such as industrial emissions and fossil fuel combustion, has been highlighted by studies focusing on severe haze episodes in East Asia (Chang et al., 2019; Liu et al., 2018a; Pan et al., 2016). The construction of high-resolution regional or global datasets has characterized the

spatiotemporal patterns of $NH_3$ emissions, with the most common datasets consisting of bottom-up inventories and top-down modeling inversion methods. Bottom-up emission inventories rely on activity data and emission factors, the latter being sensitive to assumptions about fertilizer types, local soil and meteorological properties (Bouwman et al., 2002; Søgaard et al., 2002; Xu et al.,2024). Zhang et al. (2018) evaluated discrepancies between bottom-up $NH_3$ emission inventories to exceed a factor of two, due to the uncertainties in emission factors, meteorological properties,

and agricultural statistics (Beusen et al., 2008). Crippa et al. (2018) pointed out that the uncertainty in estimated $NH_3$ emissions is largest among all pollutants in the Emissions Database for Global Atmospheric Research (EDGAR v4.3.2), with a range of variation from 186% to 294% in 2012. In contrast, the ability of satellites to measure $NH_3$ abundance combined with numerical simulations allows a better characterization of the spatial distribution and seasonality of $NH_3$ emissions, but the low signal-to-noise ratio over low-emission areas limits the accuracy of retrieval

products (Morán et al., 2016; Xu et al., 2016; Kong et al., 2019). Using a hybrid inverse modeling approach, Chen et al. (2021) optimized the $NH_3$ emission inventory NEI over the United States by combining CMAQ model simulations with constraints from the Infrared Atmospheric Sounding Interferometer (IASI) retrieval product. They found a 26% low bias of $NH_3$ emissions in the NEI, and the optimized $NH_3$ emission inventory improved the model performance of $PM_{2.5}$ mass concentration in the Midwest US, and the normalized mean bias of $NH_4^+$ and $NO_3^-$ decreased from 27%

to 22%, and 64% to 55%, respectively.

Recently, the emissions of $SO_2$ and $NO_x$ have decreased due to the implementation of related clean air policies in East Asia, North America and Europe (Zheng et al., 2018; Hand et al., 2012; Russell et al., 2012; Aas et al., 2019; Gong et al.,2024), while the emissions of $NH_3$ remained stable or have slightly increased in major agricultural regions such as China (Xu et al., 2016), the United States (Yu et al., 2018), and Europe (Fortems-Cheiney et al., 2022). The

increase in $NH_3$ emissions is associated with increasing fertilizer use and local temperatures (Warner et al., 2017; Xu et al., 2016; Skjøth and Geels, 2013; Xu et al., 2019), and such trends are expected to continue on a global scale throughout the century (IPCC, 2013). Driven by such emission trend of $SO_2$, $NO_x$, and $NH_3$, the main composition of SNA has changed in large regions with a shift from an ammonium sulfate to an ammonium nitrate formation regime (Lei et al., 2021; Jo et al., 2020; Zhou et al., 2019; Shah et al., 2018; Wang et al., 2013; Hauglustaine et al., 2014). Li

et al. (2017) evaluated that during the period 1989 to 2013, the increasing trend of sulfate ($SO_4^{2-}$) and ammonium ($NH_4^+$) mass concentration in India and China occurred at a rate >0.1 $\mu g\ m^{-3}\ yr^{-1}$, while decreasing trends were found in North America and Europe at a rate of about 0.1 $\mu g\ m^{-3}\ yr^{-1}$. The reduction of $SO_2$ and $NO_x$ results in excess $NH_3$ being released to the atmosphere because less $NH_3$ is required to neutralize $H_2SO_4$ and $HNO_3$, which in turn hinders the formation of $NH_4^+$ in the aerosol phase and increases the atmospheric $NH_3$ concentration (Liu et al., 2018b). Warner

et al. (2017) assessed that the average increase rate of $NH_3$ mass concentration from 2002 to 2016 in the United States,





Europe, and China was 2.6% yr$^{-1}$, 1.8% yr$^{-1}$, and 2.3% yr$^{-1}$, respectively. Some studies further pointed out that the increase in NH$_3$ concentration may offset the effectiveness of PM$_{2.5}$ control achieved via SO$_2$ and NO$_x$ emission reduction by promoting the formation of nitrate (NO$_3^-$) (Huang et al., 2021; Cai et al., 2017; Zhang and Geng, 2019; Fu et al., 2017).

Several studies concluded that reducing NH$_3$ emissions would be a cost-effective way to control PM$_{2.5}$ concentrations (Gu et al., 2021; Tsimpidi et al., 2007; Erisman and Schaap, 2004). However, the response of SNA to changes in its precursors is not linear (Pozzer et al., 2017; Wang et al., 2011; Wang et al., 2013; West et al., 1999), because the gas-particle partitioning of NH$_3$/NH$_4^+$ and HNO$_3$/NO$_3^-$ is influenced by several parameters, such as temperature, liquid water content, and aerosol acidity (Xu et al., 2020; Nenes et al., 2020; Guo et al., 2018). Nenes et

al. (2020) developed a conceptual framework to describe the sensitivity of particulate matter to NO$_x$ and NH$_3$ emissions, highlighting the critical influence of aerosol acidity and liquid water content on particulate matter formation. Based on sensitivity tests, Guo et al. (2018) evaluated that the response of NO$_3^-$ to NH$_3$ reduction shows an apparent decrease only when the aerosol pH falls below the value of 3.

     On the other hand, reductions in SO$_2$ and NO$_x$ emissions are expected to reduce aerosol acidity, but recent studies

revealed that aerosol acidity does not decrease as expected (Chen et al., 2019; Guo et al., 2017a; Zheng et al., 2022; Karydis et al., 2021). Aerosol acidity affects many processes involving the atmosphere and various aspects of the Earth system (Pye et al., 2020; Tilgner et al., 2021; Karydis et al., 2021) as well as human health (Dockery et al., 1993; Dockery et al., 1996; Thurston et al., 1994; Spengler et al., 1996). Weber et al. (2016) found that the acidity of PM$_{2.5}$ in the southeastern United States remained at a relatively constant level with a pH value of 0 – 2 over the past 15

years, despite a 70% reduction in SO$_4^{2-}$ concentration. A lack of aerosol acidity trend was also reported in China (Zhou et al., 2022). This is mainly caused by the buffering effect of NH$_3$/NH$_4^+$ (Chen et al., 2019; Zheng et al., 2022; Zheng et al., 2020). To investigate this, Song et al. (2019) derived an equation from the partitioning of NH$_3$, and estimated that a unit increase in pH requires a tenfold increase in NH$_3$ concentration, which is consistent with the findings of Guo et al. (2017a). The regional variation of aerosol acidity is considerable, and the pH of PM$_{2.5}$ in northern China is in the

range of 4 to 5, which is higher than in Europe and the United States (Shi et al., 2019; Shi et al., 2017; Liu et al., 2017; Guo et al., 2015; Guo et al., 2016; Guo et al., 2017b; Karydis et al., 2021). This is caused by multiple driving factors, including aerosol mass concentration and composition, NH$_3$ mass concentration, aerosol water content, and meteorological factors (Ding et al., 2019; Zhang et al., 2021a). However, the main driver for the difference in aerosol acidity remains controversial. Zheng et al. (2020) pointed out that aerosol water content is the most important factor

causing the regional variation of aerosol acidity, while Zhang et al. (2021a) emphasized the equal importance of aerosol mass concentration and chemical composition for the aerosol acidity contrasts between China and the United States.

     Almost all recent studies that discuss the response of aerosol composition and acidity to changes in NH$_3$ emission trends focus on the fine mode (e.g. PM$_{2.5}$). The size-resolved composition of SNA is not uniform (Karydis et al., 2016;

Fang et al., 2017; Karydis et al., 2011; Karydis et al., 2010; Guo et al., 2017b), and NH$_4^+$ and SO$_4^{2-}$ are mainly concentrated in the fine mode (Wang et al., 2012; Seinfeld and Pandis, 2016), while NO$_3^-$ aerosol can be formed on the surface of super-micron particles via heterogeneous chemistry (Allen et al., 2015). Furthermore, Milousis et al.



(2024) revealed that the acidity of fine-mode aerosol is more sensitive to $NH_3$ emission than coarse-mode aerosol. Reducing the $NH_3$ emissions by half, the simulated pH of fine and coarse mode aerosol decreased by up to 3 and 1.5 units, respectively. Aerosol acidity tends to decrease with increasing particle size, with pH varying up to 6 units (Craig et al.,2018; Fang et al.,2017; Bougiatioti et al.,2016). Size-resolved aerosol acidity is associated with different formation pathways (Tilgner et al.,2021; Cheng et al.,2016). Ding et al. (2019) found that the coarse-mode aerosol shifted from neutral to weakly acidic with the increase of $NO_3^-$ and $SO_4^{2-}$ during severe hazy days. Cheng et al. (2016) further pointed out that the dominant oxidant in $SO_4^{2-}$ production by $SO_2$ oxidation changes with the ambient aerosol acidity. Therefore, it is necessary to comprehensively investigate the response of size-resolved chemical composition and acidity to changes in $NH_3$ emissions.

In this study, three different $NH_3$ emission schemes are used as input to the atmospheric chemistry–climate model EMAC (ECHAM5/MESSy Atmospheric Chemistry). The three emission schemes include two bottom-up emission inventories (CAMS and CEDS_GBD), and an updated emission inventory produced following a top-down method (Luo et al., 2022). Satellite retrievals of $NH_3$ column concentrations and aerosol composition observational datasets from multiple sites around the world are used to evaluate the aerosol simulations derived from the three $NH_3$ emission schemes. We investigate the response of size-resolved aerosol SNA composition and acidity to changes in $NH_3$ emissions across three well-characterized, anthropogenically polluted regions in the Northern Hemisphere. These regions, representing a gradient from relatively less to more polluted conditions, include the United States, Europe, and the North China Plain. The examined aerosol sizes range (diameter) from sub-micron (0 – 1 μm) to super-micron (1 – 2.5 μm, 2.5 – 5 μm, and 5 – 10 μm).

## 2. Modelling description

EMAC (ECHAM5/MESSy) is a global atmospheric chemistry and climate model, which includes a number submodels describing atmospheric processes and interactions among oceans, land, and anthropogenic influences (Jöckel et al., 2016). These submodels are linked to the base model, the 5th generation European Centre Hamburg general circulation model (Roeckner et al., 2006), via the Modular Earth Submodel System (Jöckel et al., 2005). In this study, the horizontal resolution of the EMAC model is T63L31, which corresponds to a grid resolution of about 1.875° × 1.875° (Jöckel et al., 2010) and 31 vertical layers extending up to 25 km altitude. EMAC is applied for 4 years, from 2009 to 2012 with the first year used as a spin-up. The meteorological reanalysis data ERA5 (Hersbach et al., 2020), with a significantly enhanced horizontal resolution of 31km and hourly output throughout, is used in EMAC to nudge the simulation.

In EMAC, organic aerosol (OA) formation is simulated by the ORACLE module (Tsimpidi et al., 2014; Tsimpidi et al., 2016), where logarithmically spaced saturation concentration bins are used to describe the organic aerosol components based on their volatility. The aerosol microphysics and gas/aerosol partitioning are calculated by the Global Modal-aerosol eXtension (GMXe) module described by Pringle et al. (2010), which has the same microphysical core as the M7 sub-model (Vignati et al., 2004). The aerosol size distribution is treated by 7 log-normal modes, including 4 hydrophilic and 3 hydrophobic modes, covering nucleation (soluble only), Aitken, accumulation, and coarse modes (both soluble and insoluble). To determine size-resolved aerosol composition and pH, we sum the



contributions of each aerosol component, water content, and $H^+$ concentration across all GMXe modes corresponding
to a given size range. This is achieved by calculating the volume fraction of the lognormal distribution of each mode
that falls within the specified size limits. The atmospheric chemistry module MECCA (Module Efficiently Calculating
the Chemistry of the Atmosphere), which contains a comprehensive atmospheric reaction is used to calculate the gas
concentrations (Sander et al., 2019). The SEDI module is used to compute aerosol particle sedimentation (Kerkweg
et al., 2006). Dry deposition and wet deposition of gas and particle species are calculated by the DRYDEP module
(Kerkweg et al., 2006) and the SCAV module (Tost et al., 2006), respectively. The CLOUD submodel (Roeckner et
al., 2006) is used to calculate cloud properties and microphysics, utilizing the microphysical scheme of Lohmann and
Ferrachat (2010) and a physically based treatment of liquid (Karydis et al., 2017) and ice crystal (Bacer et al., 2018)
activation processes.

An advanced parameterization scheme is incorporated into the EMAC model to calculate the dust emission flux
online (Astitha et al., 2012). The scheme uses the online meteorological fields from the EMAC model, such as
temperature, pressure, relative humidity, soil moisture, and surface friction velocity, to calculate the threshold friction
velocity which is the initial step of dust production. Above the threshold friction velocity, dust emission is possible.
Following Karydis et al. (2016), the emissions of individual crustal species in this study are estimated as constant
fractions of the dust emission (Klingmüller et al., 2018). These fractional factors depend on the geological information,
which includes different dust emission sources. Karydis et al. (2021) pointed out that the crustal ions ($Ca^{2+}$, $Mg^{2+}$, $K^+$,
and $Na^+$), especially $Ca^{2+}$, have significantly contributed to maintaining the particle pH value at the level of 4.5 – 5 in
East Asia during the last decade. The importance of crustal ions in determining aerosol acidity and factors such as
liquid water content, aerosol mass concentration, and chemical composition, has been highlighted in other studies
(Zheng et al., 2020; Zhang et al., 2021a; Ding et al., 2019).

## 2.1 NH₃ emission scheme

In this study, three NH₃ emission schemes are applied in specific model simulation cases to quantify the impact of
changes in NH₃ emissions on aerosol composition and acidity (Table 1), which are divided into Bottom-up and Top-
down schemes.

**Table 1.** Setups for simulation cases.

| Simulation Case | NH₃ Emission Scheme | Equilibrium State |
|---|---|---|
| Base | CAMS-GLOB-ANT, CAMS-GLOB-AIR, GEIA, biomass burning | Stable |
| CEDS | Same as Base case, but NH₃, SO₂, and NOₓ are from CEDS-GBD | Stable |
| Top-Dep | Luo's method with the lifetime derived from deposition, GEIA-water | Stable |
| Meta | Same as Base case | Metastable |
| noNH₃ | No NH₃ emission input | Stable |

### 2.1.1 Bottom-up scheme

Bottom-up schemes are applied in the base and the CEDS cases, with different anthropogenic emission inventories.
The anthropogenic emission inventory utilized in the base case simulation is CAMS-GLOB-ANT (v4.2,
https://eccad3.sedoo.fr, abbreviated as "CAMS"), which contains 17 sectors with a spatial resolution of 0.1 × 0.1
degree and monthly temporal resolution. With the basic data of 2010 from the Emission Database for Global
Atmospheric Research (EDGAR, v4.3.2), CAMS extends the period to recent years based on the trend factors derived



from the Community Emissions Data System (Hoesly et al., 2018). Meanwhile, Hoesly et al. (2018) pointed out that there are limitations in the system, especially in the emission trends for specific sectors, and emphasized the need for more detailed data to be incorporated into regional emission inventories. The other one used in the CEDS sensitivity simulation case for aerosol precursor emissions ($NH_3$, $SO_2$, and $NO_x$) is CEDS_GBD-MAPS (Mcduffie et al., 2020),

abbreviated as "CEDS_GBD". The CEDS_GBD is developed using the Community Emissions Data System and is reported as a function of 11 anthropogenic sectors and 4 fuel categories, with a spatial resolution of 0.5 × 0.5 degrees and monthly temporal resolution. Assuming that the specific regional emission inventories are more accurate, Mcduffie et al. (2020) updated the activity data and the core scaling procedure, modified the final emission gridding and aggregation procedures, and then utilized several regional emission inventories to improve the previous version

of CEDS_GBD via the scaling procedure, which can not only reduce the discrepancy with other global emission inventories but also help to maintain the timeliness and regional accuracy of the global estimates. However, they also pointed out that the sources of uncertainty in the CEDS_GBD are similar to those in the CEDS.

Other $NH_3$ emissions include aircraft emissions from the CAMS-GLOB-AIR inventory (v1.1, https://eccad3.sedoo.fr ), land and water biological emissions from the Global Emissions Inventory Activity (GEIA)

inventory, and biomass burning emissions calculated by the BIOBURN submodel (Kaiser et al., 2012). BIOBURN determines the flux based on biomass burning emission factors and dry matter combustion rates from the Global Fire Assimilation System (GFAS), which calculates biomass burning emissions by assimilating Fire Radiative Power (FRP) observations from MODIS (Andreae, 2019).

### 2.1.2 Top-down scheme

The $NH_3$ emission inventory over land is updated by a top-down method with the constraint of IASI satellite observations (https://iasi.aeris-data.fr) developed by Luo et al. (2022). This fast top-down method updates the prior $NH_3$ emissions ($E_{NH_3,mod}$, molecule $m^{-2}$ $s^{-1}$) with a correction term positively proportional to the biases between observed ($C_{NH_3,obs}$, molecule $m^{-2}$) and simulated ($C_{NH_3,mod}$, molecule $m^{-2}$) monthly averaged $NH_3$ total column densities and inversely proportional to the $NH_3$ lifetime ($\tau_{NH_3,mod}$, s) (Eq. 1). The $\tau_{NH_3,mod}$ is calculated as the ratio

of the simulated $NH_3$ column and the sum of the simulated loss rate of the $NH_x$ family ($NH_x \equiv NH_3 + NH_4^+$) through dry and wet deposition (Eq. 2).

$$E_{NH_3} = E_{NH_3,mod} + \frac{C_{NH_3,obs} - C_{NH_3,mod}}{\tau_{NH_3,mod}} \tag{1}$$

$$\tau_{NH_3,mod} = \frac{C_{NH_3,mod}}{D_{NH_3,mod} + D_{NH_4^+,mod}} \tag{2}$$

The fast top-down method relies on the total column concentrations retrieved by IASI. According to Dammers et

al. (2019), the dominant source of uncertainty in the IASI observational product stems from the systematic bias, with the negative bias estimated between 25% to 40%, compared to site observations (Dammers et al., 2017). While this method simplifies the chemical and physical processes governing NH3, Luo et al. (2022) identified large uncertainties in regions like Central Asia and tropical Africa due to poorly constrained sources by IASI observations in these areas. Nonetheless, they demonstrated that simulations driven by the updated top-down emission inventory show better





consistency with satellite observations compared to those driven by the prior emission inventory.

### 2.1.3 Emission comparison

    Overall, the global $NH_3$ emissions used in this study in all simulation cases range from 74. Tg yr$^{-1}$ to 85 Tg yr$^{-1}$ (Table 2), which is within the reported range of the current literature from 52 Tg yr$^{-1}$ to 91 Tg yr$^{-1}$ (Schlesinger and Hartley, 1992; Dentener and Crutzen, 1994; Bouwman et al., 1997; Olivier et al., 1998; Van Aardenne et al., 2001;

Bleeker et al., 2013). The distribution of the global $NH_3$ emission flux derived from the base case and the absolute emission flux difference between the sensitivity simulation cases and the base case are shown in Figure 1. Significant regional variations in $NH_3$ emission flux are found in Figure 1 (a), with the maximum emission flux exceeding 7 g m$^{-2}$ yr$^{-1}$ in northern India, eastern China, and central Europe, all regions with the highest population density. Other emission hotspots include the eastern United States, southeastern Latin America, and central and eastern Africa. The

base case is able to capture the global $NH_3$ emission hotspots reported by Van Damme et al. (2018). Meanwhile, the minimum flux below 0.01 g m$^{-2}$ yr$^{-1}$ is located in the Antarctic and Arctic regions and the Sahara Desert, as well as in remote oceans, where there is little impact from human activities. Agricultural activities including livestock and fertilization are the main source of $NH_3$ emissions in China, India, and the United States (Liu et al., 2022; Khan et al., 2020; Van Damme et al., 2018; Sahoo et al., 2024), while soil emissions, biomass burning and domestic fires are the

main contributors to $NH_3$ emissions in central and eastern Africa (Hickman et al., 2021; Delon et al., 2012).

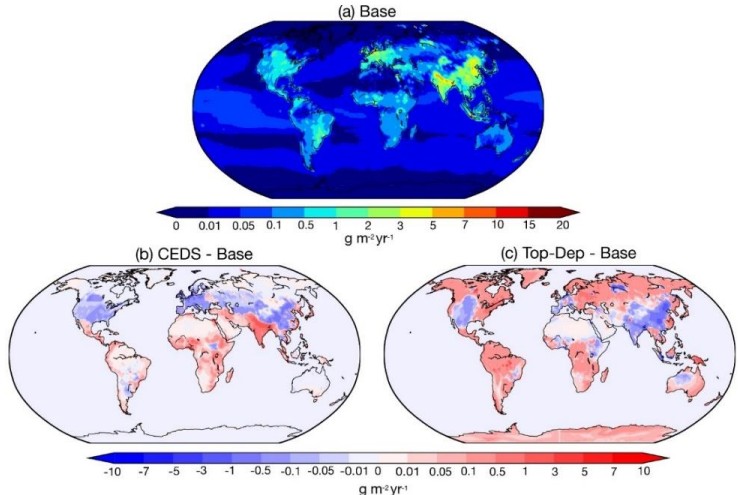

**Figure 1.** Global distribution of the annual average $NH_3$ emission flux for (a) the Base case and the absolute differences between (b) the Base and CEDS and (c) the Base and Top-Dep cases during 2010-2012.


    Compared with the base case, both the CEDS and Top-Dep cases show an increase in the global $NH_3$ emission amount, with increases of 5% and 16%, respectively. The lower $NH_3$ emission fluxes of the CEDS case are found in North America, Europe, and China except for the northeastern and southeastern coastal areas, while emission fluxes are higher throughout India (Figure 1b). Slightly higher fluxes are also found in western and eastern Africa, the





northern Middle East, and southeastern Asia. According to Mcduffie et al. (2020), the NH₃ emission flux from the Multi-resolution Emission Inventory for China (MEIC, http://www.meicmodel.org), European Monitoring and Evaluation Programme (EMEP), and US EPA are used to scale the previous emission (the basic data of 2010 from EDGAR) over mainland China, Europe, and the USA. The NH₃ emission flux from India and Africa remains the same as that of the original inventory. Constrained by the IASI satellite observation, the NH₃ emission flux of the Top-Dep

case is increased in most regions of the world, but lower emission fluxes are estimated in regions such as western North America, western and southern Europe, India, China except the northeastern and southeastern coastal areas, and western Australia. The comparison of the absolute and relative difference between the base case and the other emission schemes is summarized in Table 2.

**Table 2.** Comparison of annual NH₃ emissions (Tg yr⁻¹) across global and regional scales between the two sensitivity cases cases and the base case.

| Region | Base (Tg yr⁻¹) | CEDS | | Top-Dep | |
|---|---|---|---|---|---|
| | | diff[a] | relative diff[b] (%) | diff[a] | relative diff[b] (%) |
| Globe | 73.27 | 3.86 | 5.3 | 11.44 | 16 |
| Land | 61.14 | 3.05 | 5.0 | 11.26 | 18 |
| North America | 5.56 | -1.00 | -18 | 3.23 | 58 |
| South America | 8.20 | 0.69 | 8.4 | 4.18 | 51 |
| Europe | 6.44 | -1.50 | -23 | 1.01 | 16 |
| Middle East | 1.05 | 0.21 | 20 | -0.01 | -1.4 |
| South Asia | 10.11 | 3.79 | 38 | -0.98 | -9.7 |
| East Asia | 15.75 | -2.03 | -13 | -2.58 | -16 |

[a]: absolute difference between sensitivity cases and the base case.
[b]: relative difference between sensitivity cases and the base case.

### 2.2 ISORROPIA II

The thermodynamic equilibrium model ISORROPIA II is used to calculate the multi-phase mass transfer of the $K^+–Ca^{2+}–Mg^{2+}–NH_4^+– Na^+–SO_4^{2-}–NO_3^-–Cl^-–H_2O$ aerosol system (Nenes et al., 1998; Fountoukis and Nenes, 2007). The process of gas/aerosol partitioning is calculated in two steps (Pringle et al., 2010). In the first step, the amount of gas phase species kinetically able to condense on the aerosol within one timestep is calculated (Vignati et al., 2004). ISORROPIA II then re-distributes the mass between the gas and aerosol phase. In this study, ISORROPIA II runs in

the forward mode with the input of relative humidity, temperature, and concentration of aerosol and gas phase species. ISORROPIA II determines the subsystem set of equilibrium equations and solves the equilibrium state by the chemical potential method. It outputs the equilibrium concentration of species in gas, solid, and liquid phases by assuming that the particle phase is in the thermodynamically stable-state mode where salts precipitate once the aqueous phase becomes saturated (Fountoukis and Nenes, 2007).

Meanwhile, a sensitivity case assuming the particle phase in the thermodynamically metastable state mode is performed with the same emission scheme as the base case. In the metastable state, the aerosol may be supersaturated with respect to dissolved salts and always consists only of an aqueous phase (Fountoukis and Nenes, 2007). Karydis et al. (2021) pointed out that more acidic particles (up to 2 pH units) are derived from the metastable assumption in regions affected by high concentrations of crustal cations and consistently low relative humidity values.

According to past studies, the treatment of crustal species (e.g. $Ca^{2+}$, $K^+$, $Mg^{2+}$) in ISORROPIA II can improve model predictions (Karydis et al., 2010; Karydis et al., 2011), as both the phase partitioning of $NO_3^-$ and the



thermodynamic interaction between $NH_4^+$ and the remaining ions in the aqueous phase are significantly affected. Karydis et al. (2016) found that when these crustal species are included in the EMAC model, the increase in global $NO_3^-$ tropospheric load can be up to 44% while the global $NH_4^+$ tropospheric load decreases by 41%.

### 2.2.1 pH calculations

The pH is calculated from the negative decimal logarithm of the hydrogen ion activity,

$$pH = -log_{10}(\gamma x_{H^+}) \tag{3}$$

where $x_{H^+}$ and $\gamma$ represent the molality of hydrogen ions in the solution and the ionic activity coefficient of hydrogen, respectively. With $\gamma$ assumed to be unity, the pH value is derived using the hydrogen ion concentration in the aqueous particle phase output by ISORROPIA-II (in mol m$^{-3}$) and the aerosol water content output by GMXe (in mol mol$^{-1}$). Both hydrogen ion and aerosol water content are output every 5 hours, following Karydis et al. (2021). In addition, the temperature threshold of 269 K is set to ensure that the calculations are performed only when liquid water is present in the aerosol.

### 2.2.2 Two factors affecting pH value change

According to Equation (3), the pH value is determined by the concentrations of $H^+$ and $H_2O$. To evaluate the impact of each factor on the pH value, we independently calculate the changes in pH arising from two pathways: one driven by $H^+$ and the other by $H_2O$. The corresponding results are presented in Figures S16 and S17.

$$\Delta pH_{H_2O} = log_{10} \frac{H_2O + \Delta H_2O}{H^+} - log_{10} \frac{H_2O}{H^+} \tag{4}$$

$$\Delta pH_{H^+} = log_{10} \frac{H_2O}{H^+ + \Delta H^+} - log_{10} \frac{H_2O}{H^+} \tag{5}$$

Here, $\Delta pH_{H_2O}$ and $\Delta pH_{H^+}$ represent the changes in pH caused by variations in $H_2O$ and $H^+$ concentrations, respectively. The base case concentrations of $H_2O$ and $H^+$ are used as references, and the changes in concentration are expressed as $\Delta H_2O$ and $\Delta H^+$, corresponding to the deviations in $H_2O$ and $H^+$ from their base case levels.

### 3. Observations

Multiple observational datasets are used in this study to validate the model simulation in different regions of the world as defined by the IPCC (2023). The information of each dataset is summarized in Table 3 and the site distribution is plotted in Figure 2 and Figure S1. These datasets include satellite retrievals from the Infrared Atmospheric Sounding Interferometer (IASI; https://iasi.aeris-data.fr/NH3_IASI_A_L3_data/), and observation site networks Nationwide Nitrogen Deposition Monitoring Network (NNDMN; https://figshare.com/articles/dataset/Data_Descriptor_Xu_et_al_20181211_Scientific_data_docx/7451357/5), the European Monitoring and Evaluation Programme (EMEP; https://ebas-data.nilu.no/Default.aspx), the Central Pollution Control Board (CPCB; https://cpcb.nic.in/), the Acid Deposition Monitoring Network in East Asia (EANET; http://www.eanet.asia/product/index.html), Ammonia Monitoring Network (AMoN;



https://nadp.slh.wisc.edu/networks/ammonia-monitoring-network/), the U.S. Environmental Protection Agency (EPA; https://www.epa.gov/data), and the Clean Air Status and Trends Network (CASTNET; http://www.epa.gov/castnet).


**Table 3.** Information for each observation dataset used to validate model simulation during 2010-2012.

| Dataset | Parameters | Location |
|---------|-----------|----------|
| IASI | $NH_3$ column concentration | Globe[a] |
| NNDMN | $NH_3$, $NH_4^+$, $NO_3^-$ mass concentration | China (29 sites) |
| EMEP | $NH_3$, $NH_4^+$, $NO_3^-$, and $SO_4^{2-}$ mass concentration | Europe (25 sites for $NH_3$; 7 sites of $PM_{25}$ matrix[b]) |
| CPCB | $NH_3$ mass concentration | India (8 sites) |
| EANET | $NH_4^+$, $NO_3^-$, and $SO_4^{2-}$ mass concentration | eastern and southeastern Asia (50 sites, $PM_{25}$ matrix[b]) |
| AMoN | $NH_3$ and $NH_4^+$ mass concentration | America (21 sites) |
| CASTNET | $NH_3$, $NH_4^+$, $NO_3^-$, and $SO_4^{2-}$ mass concentration | America (79 sites, $PM_{25}$ matrix[b]) |
| EPA | $NH_4^+$, $NO_3^-$, and $SO_4^{2-}$ mass concentration | America (211 sites, $PM_{25}$ matrix[b]) |

[a]: Only use the data over land.
[b]: Measurements refer to a chemical or physical property of the total aerosol particle phase in the size fraction less than 2.5 micrometer median aerodynamic diameter.


Due to its better precision resulting from favorable thermal contrast conditions (Clarisse et al., 2009), only the morning (around 9:30 local time) overland IASI data is used in this study. The original temporal resolution of the various datasets includes hourly, three-day, daily, weekly, bi-weekly, and monthly, which we uniformly convert to monthly. The mean bias (MB), mean absolute gross error (MAGE), normalized mean bias (NMB), normalized mean error (NME), and root mean square error (RMSE) are calculated to evaluate the model performance:


$$MAGE = \frac{1}{N}\sum_{i=1}^{N}|P_i - O_i| \tag{6}$$

$$MB = \frac{1}{N}\sum_{i=1}^{N}(P_i - O_i) \tag{7}$$

$$NME = \frac{\sum_{i=1}^{N}|P_i - O_i|}{\sum_{i=1}^{N} O_i} \tag{8}$$

$$NMB = \frac{\sum_{i=1}^{N}(P_i - O_i)}{\sum_{i=1}^{N} O_i} \tag{9}$$

$$RMSE = \left[\frac{1}{N}\sum_{i=1}^{N}(P_i - O_i)^2\right]^{\frac{1}{2}} \tag{10}$$

where $P_i$ and $O_i$ represent the monthly value of model simulation and measurement, respectively. $N$ is the total number of data points used for comparison.





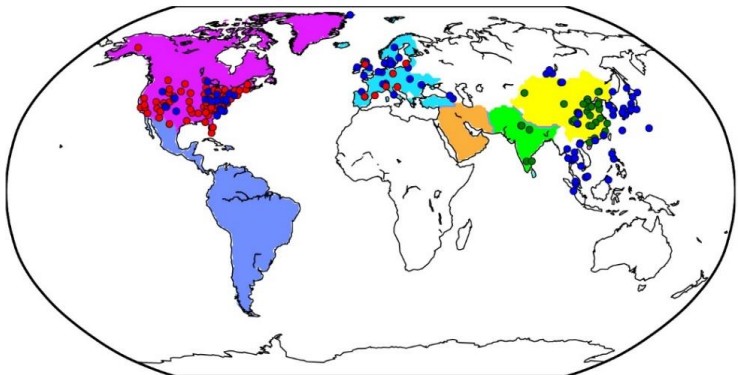

**Figure 2**. Regions and observation sites used in this study. North America, South America, Europe, Middle East, South Asia, and East Asia are marked by purple, navy blue, sky blue, orange, light green, and yellow, respectively. AMoN and CASTNET sites are represented by blue and red circles in North America (the EPA sites are shown in Figure S3). EMEP sites in Europe are shown by blue and red circles. CPCB sites are marked by green circles in South Asia, while NNDMN and EANET sites are indicated by dark green and dark blue in East Asia.

## 4. Observation evaluation

### 4.1 Aerosol comparison

The comparison between the simulation derived from the base case and the observational datasets is summarized in Tables 4 – 6. Compared with the NNDMN dataset, of which sites are mainly located in urban and rural areas of China, the base case overestimates the $NH_3$ mass concentration (NMB = 0.19), underestimates the $NH_4^+$ mass concentration (NMB = -0.41) though reproduces the $NO_3^-$ mass concentration well (NMB = -0.02). Xie et al. (2022) summarized the NMB between observed $NO_3^-$ and simulated values in China as ranging from -0.97 to 1.90 based on modelling studies in the last decade. The negligible bias of the simulated $NO_3^-$ shows the good performance of the EMAC model in this region. However, the biases in the simulation of $NH_3$ and $NH_4^+$ indicate that the $NH_3/NH_4^+$ partitioning treatment is not efficient enough or over-simplified, as less $NH_4^+$ is produced even with sufficient $NH_3$. Similarly, in Europe and North America, we obtain a positive bias of $NH_3$ mass concentration (EMEP dataset: NMB = 2.26; AMoN dataset: NMB = 0.58) and a comparably lower $NH_4^+$ mass concentration (EMEP dataset: NMB = 0.05; AMoN dataset: NMB = -0.23). On the other hand, the dry deposition of $NH_3$ in China is lower than reported from observations (NMB = -0.28; not shown), which contributes to a higher atmospheric $NH_3$ concentration.

**Table 4.** Comparison of the Base case predictions with monthly average observations from China and Europe during 2010–2012.

|  | NNDMN network Metric | | | EMEP network Metric | | | |
|---|---|---|---|---|---|---|---|
|  | $NH_3$ | $NH_4^+$ | $NO_3^-$ | $NH_3$ | $NH_4^+$ | $NO_3^-$ | $SO_4^{2-}$ |
| Observed (µg m⁻³) | 7.68 | 7.45 | 11.92 | 1.16 | 1.09 | 1.29 | 2.11 |
| Simulated (µg m⁻³) | 9.14 | 4.40 | 11.70 | 3.79 | 1.15 | 3.33 | 1.24 |
| MAGE (µg m⁻³) | 5.27 | 4.21 | 6.00 | 2.78 | 0.60 | 2.38 | 0.98 |
| MB (µg m⁻³) | 1.46 | -3.05 | -0.22 | 2.63 | 0.06 | 2.04 | -0.88 |
| NME | 0.69 | 0.56 | 0.50 | 2.40 | 0.55 | 1.84 | 0.46 |
| NMB | 0.19 | -0.41 | -0.02 | 2.26 | 0.05 | 1.58 | -0.41 |
| RMSE (µg m⁻³) | 7.43 | 6.55 | 9.07 | 4.03 | 0.97 | 2.95 | 1.63 |
| Number of comparisons | 765 | 765 | 765 | 832 | 249 | 320 | 320 |





**Table 5.** Comparison of the Base case predictions with monthly average observations from India and eastern Asia ($PM_{2.5}$ matrix) during 2010–2012.

| | CPCB network Metric | EANET network Metric | | |
|---|---|---|---|---|
| | $NH_3$ | $NH_4^+$ | $NO_3^-$ | $SO_4^{2-}$ |
| Observed ($\mu g\ m^{-3}$) | 30.07 | 0.84 | 1.22 | 2.95 |
| Simulated ($\mu g\ m^{-3}$) | 22.30 | 0.83 | 2.09 | 1.52 |
| MAGE ($\mu g\ m^{-3}$) | 21.25 | 0.55 | 1.58 | 1.74 |
| MB ($\mu g\ m^{-3}$) | -7.78 | -0.02 | 0.87 | -1.43 |
| NME | 0.71 | 0.65 | 1.29 | 0.59 |
| NMB | -0.26 | -0.02 | 0.71 | -0.49 |
| RMSE ($\mu g\ m^{-3}$) | 30.57 | 1.11 | 2.58 | 2.48 |
| Number of comparisons | 137 | 908 | 916 | 886 |

**Table 6.** Comparison of the Base case predictions with monthly average observations from America ($PM_{2.5}$ matrix) during 2010–2012.

| | AMoN network Metric | | CASTNET network Metric | | | EPA network Metric | | |
|---|---|---|---|---|---|---|---|---|
| | $NH_3$ | $NH_4^+$ | $NH_4^+$ | $NO_3^-$ | $SO_4^{2-}$ | $NH_4^+$ | $NO_3^-$ | $SO_4^{2-}$ |
| Observed ($\mu g\ m^{-3}$) | 1.20 | 1.27 | 0.69 | 0.74 | 1.81 | 0.83 | 1.24 | 1.97 |
| Simulated ($\mu g\ m^{-3}$) | 1.89 | 0.99 | 0.90 | 1.96 | 1.34 | 1.02 | 2.22 | 1.44 |
| MAGE ($\mu g\ m^{-3}$) | 1.16 | 1.02 | 0.34 | 1.33 | 0.64 | 0.42 | 1.30 | 0.65 |
| MB ($\mu g\ m^{-3}$) | 0.69 | -0.29 | 0.21 | 1.22 | -0.46 | 0.19 | 0.98 | -0.53 |
| NME | 0.97 | 0.80 | 0.49 | 1.80 | 0.36 | 0.51 | 1.05 | 0.33 |
| NMB | 0.58 | -0.23 | 0.30 | 1.65 | -0.26 | 0.23 | 0.79 | -0.27 |
| RMSE ($\mu g\ m^{-3}$) | 1.64 | 1.43 | 0.45 | 1.71 | 0.86 | 0.60 | 1.76 | 0.90 |
| Number of comparisons | 553 | 552 | 2825 | 2825 | 2825 | 5085 | 5392 | 5429 |

In East and Southeast Asia, the mass concentration of $NH_4^+$ is well reproduced (NMB = -0.02), while high and low discrepancies are found in the mass concentrations of $NO_3^-$ and $SO_4^{2-}$ (NMB = 0.71 and -0.49, respectively). Similar results were also found in Europe, with agreement for $NH_4^+$ (NMB = 0.05) but an overestimation of the $NO_3^-$ mass concentration (NMB = 1.58) and underestimated $SO_4^{2-}$ (NMB = -0.41). The positive bias of the simulated $NO_3^-$ is reported by many studies (Xie et al., 2022; Heald et al., 2012; Colette et al., 2011; Bian et al., 2017; Pozzer et al., 2022). The negative bias of $SO_4^{2-}$ is considered a reason for the positive bias of $NO_3^-$, regarding the thermodynamic equilibrium between $NH_4^+$, $SO_4^-$, and $NO_3^-$. The discrepancy in $SO_4^-$ and $NO_3^-$ is also due to the missing heterogeneous oxidation reactions for $SO_2$ and $NO_x$ in the model. Several studies have concluded that adding multiphase chemistry can significantly improve the model performance (Zheng et al., 2015; Zhang et al., 2021b). Cheng et al. (2016) and Zheng et al. (2024) pointed out that the multiphase reactions act as an important $SO_4^-$ source in haze pollution, while Guo et al. (2017a) argued that the multiphase reactions are not likely limited by the required alkaline environment.

In North America, the base case reproduces the mass concentration of $SO_4^{2-}$ (NMB = -0.26), but overpredicts the mass concentrations of $NH_4^+$ and $NO_3^-$ (NMB = 0.23 and 0.79, respectively), which is in in line with the findings of Walker et al. (2012). The excess $NH_4^+$ promotes the formation of $NO_3^-$, and the uncertain uptake coefficient of $N_2O_5$ used in models may contribute to more $NO_3^-$ (Walker et al., 2012). The highest $NH_3$ mass concentration is found in India, especially in the Indo-Gangetic Plain. Our model basically captures the "hot spot", with a slight negative bias (NMB = -0.26). However, the scarce observation sites and the lack of observed $NH_4^+$, $NO_3^-$ and $SO_4^{2-}$ hinder further evaluation of the model performance.



**4.2 pH value comparison**

Due to the lack of direct measurement of aerosol acidity, we collect the $PM_{2.5}$ pH value from related studies to compare with our model simulation in Table 7. These pH values are calculated using the thermodynamic equilibrium model ISORROPIA or E-AIM with input from observational datasets. Compared with Guo et al. (2017b), our simulated pH value from the base case is higher in the western USA (4.3 vs 2.7), but the value from the Meta case is much closer (2.6). Both the base and Meta cases predict the same aerosol water content, and the high $Ca^{2+}$ concentration from the Great Basin Desert leading to the precipitation of $CaSO_4$ is the main reason for the lower aerosol acidity in the base case (Karydis et al., 2021). It is worth noting that the effect of $Ca^{2+}$ on aerosol acidity was not considered by Guo et al. (2017b). In Europe, although the good agreement between our simulated pH and the result of Guo et al. (2018) (both are 3.9) supports the model simulations, we note the overestimation of simulated alkaline species (sum of $NH_4^+$ and $NH_3$, NMB = 17%) along with underestimation of acidic species (sum of $NO_3^-$ and $HNO_3$, NMB = -57%; $SO_4^{2-}$, NMB = -51%) in December emphasizes the improvement of the aerosol representation. In northern China, the pH value calculated by Wang et al. (2016) was more than 2 units higher (4.4 vs 6.7), and we suggest that the low bias is due to the underestimation of the mass concentration of cations (e.g. $Ca^{2+}$, $Mg^{2+}$) in $PM_{2.5}$, as the mass concentrations of $NH_4^+$, $SO_4^{2-}$, and $NO_3^-$ are in the observational ranges of Wang et al. (2016).

**Table 7.** Simulated pH value of $PM_{2.5}$ at single points compared with literature.

| Location | Latitude | Longitude | Time Period | Method used | Field derived mean pH | Base | Meta | Reference |
|---|---|---|---|---|---|---|---|---|
| Egbert, ON, Canada | 44.23 | -79.78 | Jul-Sep, 2012 | E-AIM | 2.1 | 3.77 | 1.57 | Murphy et al. (2017) |
| Pasadena,CA, USA | 34.14 | -118.12 | Jun, 2010 | ISORROPIA | 2.7 | 4.26 | 2.58 | Guo et al. (2017) |
| Sao Paulo, Brazil | -23.55 | -46.63 | Aug-Sep, 2012 | E-AIM | 4.8 | 3.85 | 3.34 | Vieira-Filho et al. (2016) |
| Cabauw, Netherland | 51.97 | 4.93 | Dec-Feb, 2012 | ISORROPIA | 3.9 | 3.91 | 3.58 | Guo et al. (2018) |
| Xi'an, China | 34.23 | 108.89 | Nov-Dec, 2012 | ISORROPIA | 6.7 | 4.41 | 3.20 | Wang et al. (2016) |

Note: table extracted in part from Karydis et al. (2021).

**5. Secondary inorganic aerosol composition**

**5.1 Size-resolved composition**

The regional mass fractions of size-resolved inorganic aerosol components ($NH_4^+$, $SO_4^{2-}$, and $NO_3^-$) are presented as bar charts in Figure 3, while global distribution maps of their size-resolved mass concentrations are shown in Figure S2, both the simulation results are derived from base case. To assess ammonia neutralization of sulfuric and nitric acids, we applied the chemical domain framework defined by Ge et al. (2022) based on SNA molar concentrations in $PM_{10}$ with a threshold of >1 µg m⁻³. The four chemical domains, illustrated in Figure 4, are defined as follows: *"$SO_4^{2-}$ very rich"* (totNH₃/ totSO₄ < 1, totNH₃: sum of $NH_3$ and $NH_4^+$, totSO₄: sum of $SO_4^{2-}$ and $HSO_4^-$), *"$SO_4^{2-}$ rich"* (totNH₃/ totSO₄ between 1 and 2), *"$NO_3^-$ rich"* (free NH₃/ totNO₃ between 0 and 1, free NH₃: totNH₃ minus double totSO₄,





totNO₃: sum of $NO_3^-$ and $HNO_3$), and *"NH₃ very rich"* (free $NH_3$/ totNO₃ > 1). Figure S3 shows the ratios used to
define these domains.

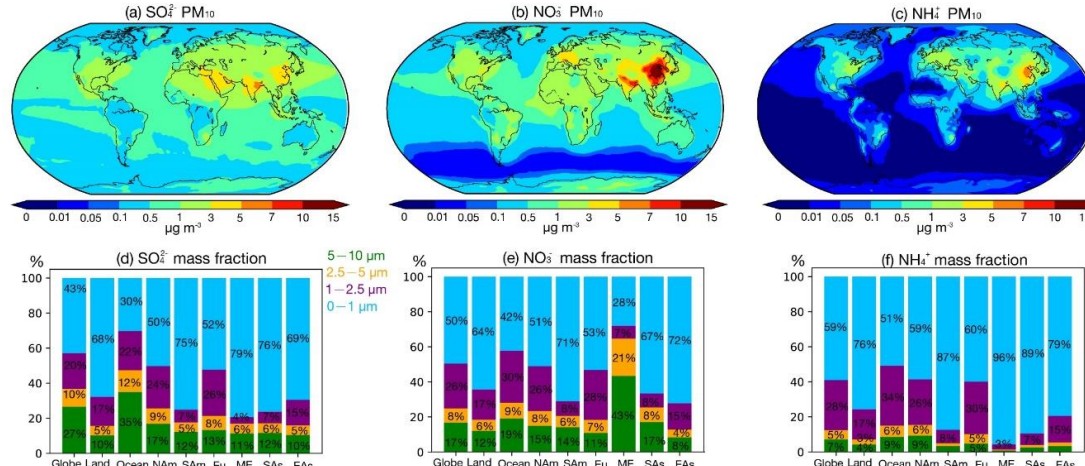

**Figure 3.** (a) – (c) global distribution of $SO_4^{2-}$, $NO_3^-$, and $NH_4^+$ mass concentration in the size range of 0 – 10 μm, (d) – (f) bar
plots for mean mass fractions of size-resolved $NH_4^+$, $SO_4^{2-}$, and $NO_3^-$ over globe, land, ocean and regions (marked in Figure 2), the
size range of 5 – 10 μm, 2.5– 5 μm, 1– 2.5 μm, and 0 – 1 μm are marked by green, orange, purple, and blue, respectively (NAm:
North America; SAm: South America; Eu: Europe; ME: Middle East; SAs: South Asia; EAs: East Asia).

Regions with low inorganic aerosol concentrations (<1 μg m⁻³) are found in Southern Hemisphere oceans and
remote areas such as the North Pole and South America, including the Amazon Basin. These areas represent relatively
pristine baselines for evaluating anthropogenic impacts (Andreae et al., 1990; Andreae, 2007). In low northern latitude
oceans, the *"SO₄²⁻ very rich"* and *"SO₄²⁻ rich"* domains dominate, characterized by $NH_3$ fully converted to $NH_4^+$ but
incomplete neutralization of sulfuric acid. This pattern reflects low $NH_3$ emissions over oceans (Figure 1) and the
contribution of biogenic dimethyl sulfide (DMS) to marine $SO_4^{2-}$ (Fiddes et al., 2018; Jackson et al., 2020). Additional
$NO_x$ and $SO_2$ from shipping contribute to marine $SO_4^{2-}$ and $NO_3^-$ (Wang et al., 2023; Burgard and Bria, 2016). Average
$PM_{10}$ SNA concentrations are lower over oceans compared to land ($NH_4^+$: 0.06 μg m⁻³ vs. 0.36 μg m⁻³; $NO_3^-$: 0.37 μg
m⁻³ vs. 1.24 μg m⁻³; $SO_4^{2-}$: 0.60 μg m⁻³ vs. 0.95 μg m⁻³). Marine $SO_4^{2-}$ and $NO_3^-$ are predominantly in the super-micron
mode, with mass fractions of 70% and 58%, respectively. Coastal areas exhibit a *"NO₃⁻ rich"* domain due to
continental outflows (Figure S2) (Prospero et al., 1985), consistent with prior findings that marine aerosol in the super-
micron mode primarily comprises inorganic salts, including sea salt, non-sea salt sulfate, and nitrate, while organic
matter is concentrated in the sub-micron range (Russell et al., 2023; Cavalli et al., 2004).

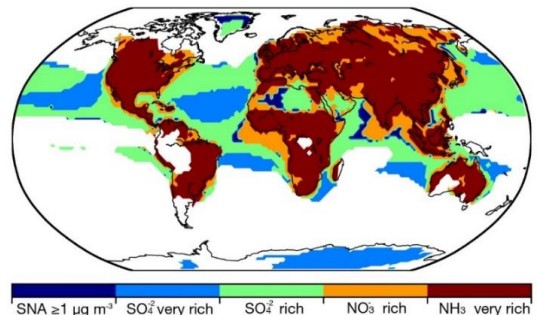

**Figure 4.** Distribution of ammonia neutralization state of sulfuric and nitric acids, based on SNA molar concentrations in $PM_{10}$ with a threshold of $\geq 1$ µg m$^{-3}$, where the SNA mass concentration less than the threshold is measured by blank zone. The "$SO_4^{2-}$ very rich" represents the ratio of totNH3 to totSO4 less than 1 (totNH3: sum of NH3 and NH4$^+$, totSO4: sum of SO4$^{2-}$ and HSO4$^-$), the "$SO_4^{2-}$ rich" represents the ratio of totNH3 to totSO4$^{2-}$ between 1 and 2, the "$NO_3^-$ rich" represents the ratio of free NH3 to totNO3 between 0 and 1 (free NH3: totNH3 minus double totSO4, totNO3: sum of NO3$^-$ and HNO3), and the "$NH_3$ very rich" represents the ratio of the free NH3 to totNO3 over 1.

On land, except for areas such as northern Russia, central Africa, and the Arabian Peninsula, the aerosol typically falls within the *"NH₃ very rich"* domain. In this domain, $SO_4^{2-}$ is fully neutralized by $NH_3$, with sufficient $NH_3$ available to additionally neutralize $NO_3^-$, making $NO_3^-$ the limiting factor in $NH_4NO_3$ formation. More than 60% of SNA mass is in the sub-micron mode on land, while the super-micron modes (i.e. 1–2.5 µm, 2.5–5 µm and 5–10 µm) accounts for a smaller fraction (7% for $NH_4^+$, 15% for $SO_4^{2-}$, and 18% for $NO_3^-$). In polluted regions such as East and South Asia, $PM_{10}$ SNA concentrations are three times higher than the global land average. For example, East Asia shows $NH_4^+$: 1.88 µg m$^{-3}$, $NO_3^-$: 5.31 µg m$^{-3}$, and $SO_4^{2-}$: 2.29 µg m$^{-3}$, while South Asia records $NH_4^+$: 1.58 µg m$^{-3}$, $NO_3^-$: 3.68 µg m$^{-3}$, and $SO_4^{2-}$: 3.18 µg m$^{-3}$. Free $NH_3$ is abundant in regions like southern North America, Europe, South Asia, and East Asia, with mean free $NH_3$/total $NO_3^-$ ratios of 2.11, 3.77, 5.30, and 3.78, respectively. Over 75% of SNA mass in these regions is concentrated in the 0–1 µm and 1–2.5 µm size ranges. The $NH_3$ surplus reflects recent trends in precursor emissions, with stable or increasing $NH_3$ emissions contrasting with declining $SO_2$ and $NO_x$ emissions (Zheng et al., 2018; Hand et al., 2012; Russell et al., 2012). In the Middle East, particularly the Arabian Peninsula, aerosol is dominated by desert dust with negligible $NH_3$ emissions. The *"$SO_4^{2-}$ rich"* and *"$NO_3^-$ rich"* domains predominate, where $NH_3$ levels are insufficient to neutralize acidic components fully, limiting $NH_4NO_3$ formation. Mean concentrations of $NH_4^+$, $NO_3^-$, and $SO_4^{2-}$ in this region are 0.72 µg m$^{-3}$, 2.07 µg m$^{-3}$, and 3.19 µg m$^{-3}$, respectively. Over 70% of $NO_3^-$ resides in the super-micron modes due to interactions with sea salt and crustal dust, which shift $NO_3^-$ from sub-micron to super-micron modes (Chen et al., 2020; Koçak et al., 2007; Karydis et al., 2016). Sub-micron $NH_4^+$ and $SO_4^{2-}$ remain dominant, accounting for 96% and 79% of their respective fractions, consistent with Osipov et al. (2022), who identified anthropogenic sources as the primary contributors to fine particles in the region.

## 5.2 NH₃ column concentration

Figure 5 (a) shows the global distribution of $NH_3$ column concentration. The global area-weight mean value of $NH_3$ column concentration is 0.80 mg m$^{-2}$, with the highest value up to 30 mg m$^{-2}$ in the Indo-Gangetic Plain of India and the lowest value of less than 0.01 mg m$^{-2}$ in the remote oceans of the Southern Hemisphere and the South Pole.



Compared with previous studies that investigated NH₃ column concentrations based on satellite retrievals (Van Damme et al., 2014; Van Damme et al., 2015; Zhou et al., 2024), our results can capture the distribution of NH₃
hotspots worldwide, including the Indo-Gangetic Plain, the North China Plain, and West Africa and Amazonia, where biomass burning is dominant (Van Damme et al., 2018). However, Van Damme et al. (2018) pointed out that two-thirds of the NH₃ emission hotspots are underestimated by at least one order of magnitude in the NH₃ emission inventory EDGAR (CAMS-GLOB-ANT used in this study is derived from EDGAR, see **2.1.1**). Given such an underestimation in the current NH₃ emission inventory, we further improve the emission by applying a new inventory
and updating the current inventory using a top-down method introduced in Section 2.1. The simulation results are discussed in Section 7.

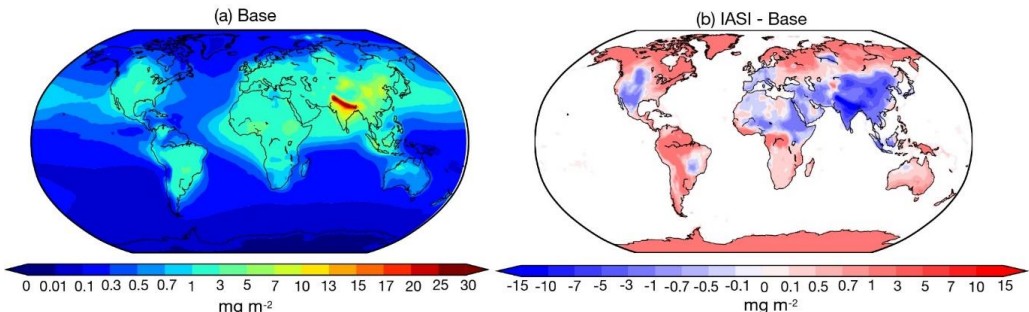

**Figure 5.** Global distribution of (a) the average NH₃ column concentration (mg m⁻²) for the Base Case, and (b) the absolute
difference between IASI satellite retrieval and Base case, from 2010 to 2012.

Our simulated NH₃ column concentrations show good agreement with the IASI satellite observations (Figure 5b), with global land mean values at a comparable level of 1.66 mg m⁻². However, regional biases can be significant, with an NMB of -0.32. The model overestimates NH₃ column concentrations over regions such as India, China, western
America, and northeastern Africa, while it has a negative bias in high-latitude regions. It is also important to note that biases in the IASI satellite products have been identified in previous studies (Dammers et al., 2017; Dammers et al., 2019), more details are discussed in Section **2.1.2**.

### 5.3 Global NH₃ budget

The global budget for NH₃ and NH₄⁺ is summarized in Table 8. Given a global NH₃ emission input of 74.37 Tg
475    yr⁻¹, the global burden and lifetime derived from the base case simulation are 0.41 Tg and 2.01 days, respectively; for NH₄⁺, the global burden and lifetime are 0.34 Tg and 3.46 days, respectively. Based on the simulations of nine models, Bian et al. (2017) assessed that given the average NH₃ emission input of 76.38 Tg yr⁻¹, the average global burden and lifetime for NH₃ and NH₄⁺ are 0.20 Tg and 0.72 days, and 0.32 Tg and 4.3 days, respectively. Compared to Bian et al. (2017), our study uses the same NH₃ emission input; however, the global burden and lifetime of NH₃ derived from
the base case are larger. This potential overestimation may be attributed to the wet deposition scheme used in EMAC. The scavenging scheme (SCAV) accounts for pH adjustments in NH₃ dissolution. More specifically, the EMAC model implicitly determines the effective Henry's law constant by solving a system of coupled ordinary differential equations, explicitly representing liquid-phase processes in clouds and raindrops, including dissociation, acid–base





equilibria, redox reactions, and photolysis (Tost et al., 2006). This approach ensures a comprehensive calculation of total wet deposition for $NH_4^+$ and $NH_3$. Notably, the $NH_3$ burdens simulated in the AeroCom model intercomparison by Bian et al. (2017) exhibit significant variability, spanning a factor of 17. This wide range underscores the sensitivity of atmospheric $NH_4^+$ and $NH_3$ burdens and lifetimes to model domain definitions, deposition pathways, and $NH_3$ chemical processes across different models (Ge et al., 2022).

**Table 8.** Atmospheric budget of $NH_3$, $NH_4^+$, and $NH_x$ ($NH_3 + NH_4^+$).

| Simulation Case | Specie | Emission (Tg yr$^{-1}$) | Burden (Tg) | Dry Deposition (Tg yr$^{-1}$) | Wet Deposition (Tg yr$^{-1}$) | Lifetime[a] (day) |
|---|---|---|---|---|---|---|
| Base | $NH_3$ | 73.27 | 0.41 | 28.22 | - | 2.01 |
| CEDS | | 77.13 | 0.43 | 29.90 | - | 2.01 |
| Top-Dep | | 84.71 | 0.41 | 34.36 | - | 1.77 |
| Base | $NH_4^+$ | - | 0.34 | 1.16 | 34.69 | 3.46 |
| CEDS | | - | 0.35 | 1.25 | 35.77 | 3.45 |
| Top-Dep | | - | 0.35 | 1.52 | 36.19 | 3.39 |
| Base | $NH_x$ | - | 0.75 | 29.38 | 34.69 | 3.68 (4.27)[b] |
| CEDS | | - | 0.78 | 31.15 | 35.77 | 3.66 (4.25) |
| Top-Dep | | - | 0.76 | 35.90 | 36.21 | 3.27 (3.85) |

[a]: $NH_3$ lifetime = Burden/Emission; $NH_4^+$ lifetime = Burden/(Dry Deposition + Wet Deposition);
[b]: $NH_x$ lifetime = $NH_x$ Burden/ $NH_3$ Emission; in the parentheses, $NH_x$ lifetime = $NH_x$ Burden/ (Dry Deposition + Wet Deposition);

Considering the SCAV scavenging scheme, we further calculate the global budget for $NH_x$ ($NH_3 + NH_4^+$) in Table 8. The global burden of $NH_x$ is 0.75 Tg. Wet and dry deposition contribute almost equally to the sink accounting for 54% and 46%, respectively. The lifetime of $NH_x$ is 3.68 – 4.27 days, depending on the calculation method. Ge et al. (2022) estimated a global budget for $NH_x$, with an input $NH_3$ emission of 64.48 Tg yr$^{-1}$. They calculated the global burden of $NH_x$ to be 0.91 Tg, with a lifetime of 4.9-5.2 days and its wet and dry deposition contributing equally to the sink (i.e., 51% and 49% of total deposition, respectively).

## 6. Aerosol acidity

Figure 6 illustrates the global distributions of size-resolved aerosol pH, with regional averages summarized in Table 9. Aerosol pH exhibits marked spatial variations, influenced by land-sea contrasts and regional sources. Over land in the Northern Hemisphere, excluding deserts, aerosols are generally acidic, whereas marine aerosols are alkaline due to sea salt influence. However, high-latitude marine aerosols are more acidic due to the low liquid water content and hydrogen ion outflow from land. The average pH values for land-based aerosols are 4.2 (5–10 μm), 4.2 (2.5–5 μm), 3.9 (1–2.5 μm), and 4.3 (0–1 μm). In contrast, marine aerosol pH values are 6.2 (5–10 μm), 5.8 (2.5–5 μm), 5.1 (1–2.5 μm), and 5.1 (0–1 μm). The deserts of North Africa, the Middle East, and the Gobi maintain the highest pH values (>7) across all size ranges, driven by non-volatile cations (e.g., $Ca^{2+}$, $Mg^{2+}$) that neutralize acidic components and enhance water uptake.

Regionally, higher pH values (5–5.7) in the Middle East are attributed to airborne dust, while coastal areas like the western Arabian Gulf have lower aerosol pH (<3 in 0–1 μm) due to elevated sulfate concentrations. In South Asia, abundant $NH_3$ emissions keep pH values between 5.4 and 4.9 despite high $SO_2$ and $NO_x$ emissions. East Asia shows a distinct pH gradient, with desert regions in the northwest reaching pH >7 across all sizes, and southeast coastal areas exhibiting low pH (2–4), linked to significant $SO_2$ emissions and sulfate formation. Europe (pH 3.7–4.1) and North




America (pH 3.2–3.6) demonstrate moderate acidity, with the western USA exhibiting higher values (4–6) due to desert influences. In South America, pH ranges from 3.9 to 4.5, with coastal regions exceeding 6 due to sea salt.

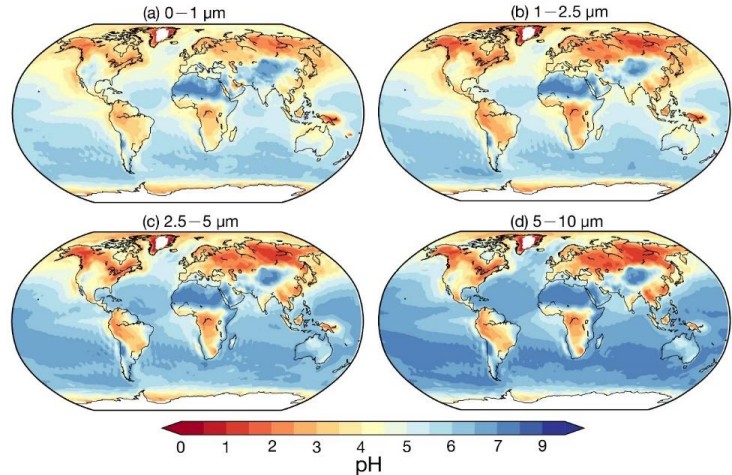

**Figure 6.** Global distribution of surface average aerosol pH values in the size ranges: (a) 0 – 1 µm, (b) 1 – 2.5 µm, (c) 2.5 – 5 µm, (d) 5 – 10 µm, derived from Base case from 2010 to 2012.

An increasing trend in pH is observed from North America to Europe and East Asia, consistent with prior studies (Zhang et al., 2021a; Ding et al., 2019; Guo et al., 2017a). Aerosol alkalinity is driven by $NH_4^+$ and non-volatile cations, which neutralize $SO_4^{2-}$ and $NO_3^-$, while water-soluble ions (WSIs) enhance liquid water uptake. East Asia exhibits the highest pH among regions, facilitated by relatively lower $SO_4^{2-}$, abundant $NH_4^+$ and WSIs, and significant $NO_3^-$ and non-volatile cations in the coarse modes. These chemical properties outweigh the influence of meteorological effects such as differences in temperature and humidity.

**Table 9.** Size-resolved pH values across the globe and regions from simulation cases.

| Region | 0 – 1 µm | | | 1 – 2.5 µm | | | 2.5 – 5 µm | | | 5 – 10 µm | | |
|---|---|---|---|---|---|---|---|---|---|---|---|---|
| | Base[a] | Meta[a] | noNH3[a] | Base[a] | Meta[a] | noNH3[a] | Base[a] | Meta[a] | noNH3[a] | Base[a] | Meta[a] | noNH3[a] |
| Globe | 4.9 | 4.19 | 3.28 | 4.83 | 4.3 | 4.09 | 5.38 | 5 | 4.82 | 5.75 | 5.47 | 5.46 |
| Land | 4.29 | 2.96 | 1.06 | 3.94 | 2.9 | 1.71 | 4.19 | 3.36 | 2.51 | 4.23 | 3.54 | 3.11 |
| Ocean | 5.1 | 4.58 | 3.99 | 5.11 | 4.75 | 4.86 | 5.76 | 5.52 | 5.56 | 6.24 | 6.08 | 6.21 |
| North America | 3.62 | 2.63 | 1.01 | 3.16 | 2.38 | 1.17 | 3.31 | 2.63 | 1.83 | 3.38 | 2.81 | 2.34 |
| South America | 4.11 | 3.25 | 1.46 | 3.85 | 3.26 | 2.26 | 4.4 | 4 | 3.25 | 4.52 | 4.22 | 3.84 |
| Europe | 4.09 | 2.97 | 0.84 | 3.65 | 2.8 | 1.27 | 3.69 | 2.92 | 1.91 | 3.75 | 3.08 | 2.67 |
| Middle East | 5.02 | 1.94 | -0.17 | 5.25 | 2.08 | 0.51 | 5.47 | 3.59 | 1.86 | 5.66 | 4.18 | 3.85 |
| South Asia | 5.42 | 2.8 | -0.2 | 4.86 | 2.33 | 0.43 | 5 | 2.73 | 1.37 | 5.04 | 2.88 | 2.16 |
| East Asia | 5.15 | 3.23 | 0.5 | 4.65 | 3.32 | 1.16 | 4.56 | 3.24 | 1.76 | 4.54 | 3.26 | 2.3 |

[a]: average pH value.



Contrary to previous findings (e.g. Kakavas et al., 2021; Ding et al., 2019), which suggest decreasing aerosol acidity with increasing particle size, pH in the base case (stable state) for 0–1 μm exceeds that of 1–2.5 μm over many regions, excluding oceans and the Middle East. This is examined through three perspectives. First, a sensitivity simulation (Meta case) assuming metastable conditions revealed lower pH across all sizes, with the greatest reductions in regions with high mineral cations and/or low relative humidity (RH), such as South Asia and the Middle East. In the metastable state, all anions remain aqueous, resulting in more acidic aerosols. While the Meta case explains some discrepancies, pH in the 0–1 μm size range remains higher relative to 1–2.5 μm in several regions (Table 5). Another sensitivity simulation, removing $NH_3$ emissions (noNH$_3$ case), significantly reduced pH across all sizes over land, especially in $NH_3$-rich regions like South Asia and the Middle East, where 0–1 μm pH dropped below zero. This indicates increased pH sensitivity in the fine size ranges (i.e., 0–1 μm and 1–2.5 μm), particularly in regions where $NH_3$ availability is abundant. Excluding $NH_3$, results in a consistent pH increase with particle size across all regions. Finally, a comparison with observed size-resolved mass concentrations from the EMEP network revealed an underestimation of acidic components ($SO_4^{2-}$ and $NO_3^-$) and an overestimation of alkaline components in the 0–1 μm size range. This imbalance contributes to the anomalously high simulated pH in 0–1 μm. For instance, the observed $SO_4^{2-}$, $NO_3^-$ and $NH_4^+$ in 0–1 μm accounts for 91%, 59% and 67% of 0–2.5 μm at the Montseny site (41.78 º N, 2.35º E) during 2010 – 2012 (observed $NH_4^+$ only available in 2010), respectively, compared to the accounting of 79%, 54% and 77% in simulation, highlighting a need for improved representation of aerosol composition in fine modes.

This study calculates size-resolved aerosol pH based on a log-normal size distribution, differing from other approaches. For instance, Kakavas et al. (2021) used the PMCAMx model, which employs a sectional approach to track aerosol mass and composition across 10 size bins ranging from 40 nm to 40 μm. The first five bins represent the $PM_1$ fraction (0.04–1.25 μm), while three bins cover sizes up to 10 μm. In addition, previous studies on size-resolved aerosol acidity have certain limitations. For example, a laboratory study observed increasing aerosol acidity with decreasing particle size, but this trend only holds for pH below 2 (Craig et al., 2018). Similarly, a field study reached the same conclusion but lacked measurements of key gas-phase species (Fang et al., 2017).

## 7. Emission Sensitivity Analysis

The formation of secondary inorganic aerosols is strongly linked to $NH_3$ emissions (Wu et al., 2016; Chen et al., 2019; Liang et al., 2024). Xu et al. (2020) and Wang et al. (2015) have highlighted the critical role of gas-particle partitioning of $NH_3/NH_4^+$ in SNA formation, which is influenced by factors such as temperature, aerosol water content, and aerosol acidity (Nenes et al., 2020). Notably, the $NH_3/NH_4^+$ partitioning process buffers aerosol acidity, maintaining stability even amid shifts in acidic species like $NO_3^-$ and $SO_4^{2-}$ (Karydis et al., 2021). These findings suggest that the effects of $NH_3$ emissions on SNA formation and aerosol acidity remain a subject of debate (Weber et al., 2016; Zheng et al., 2022; Fu et al., 2017; Zou et al., 2024). However, most studies have focused on fine particles ($PM_{2.5}$ or $PM_1$), polluted regions (e.g., northern China), and haze episodes (Ge et al., 2019; Gao et al., 2016). Modeling studies often evaluate the effects of $NH_3$ emission changes by uniformly altering emission levels within each grid (Pozzer et al., 2017; Fu et al., 2017). While such approaches provide valuable insights, they may lack feasibility as $NH_3$ abundance correlates with population density, making uniform changes less representative of real-world





scenarios. To address this, we compare the effects of two $NH_3$ emission schemes CEDS and Top-Dep on size-resolved SNA mass concentration and aerosol acidity relative to a base case.

The $NH_3$ mass concentration serves as a proxy for $NH_3$ emissions, and differences in $NH_3$ mass concentrations between the scenarios and the base case (Figure S6) align with corresponding emission differences (Figure 1). Figures 7 and 8 show the responses of size-resolved $NH_4^+$, $NO_3^-$, and $SO_4^{2-}$ mass concentrations to $NH_3$ emission changes in the CEDS and Top-Dep cases, respectively. Similarly, Figure 9 illustrates the size-resolved aerosol acidity responses for the two cases. Additional details are provided in supplementary figures: Figures S7–S9 (CEDS case) and Figures

S11–S13 (Top-Dep case) depict the size-resolved responses of $NH_4^+$, $NO_3^-$, and $SO_4^{2-}$ mass concentrations, while Figures S10 and S14 highlight changes in size-resolved aerosol acidity.

     The regional gas-particle partitioning ratios for $NH_3/NH_4^+$ ($\varepsilon(NH_4^+) = NH_4^+/(NH_4^+ + NH_3)$) and $NO_3^-/HNO_3$ ($\varepsilon(NO_3^-) = NO_3^-/(NO_3^- + HNO_3)$) are shown in Figure S15. Regional emission amounts of $NO_x$ and $SO_2$ from the CAMS and CEDS_GBD inventories are detailed in Table S1.

**7.1 Size-Resolved SNA Response**

     Regarding the atmospheric budget of $NH_3$ (Table 8), a modest increase in global $NH_3$ emissions (CEDS case) slightly raises the burden and deposition of $NH_3$ and $NH_4^+$, while the lifetime remains unchanged. In contrast, the larger emission increase in the Top-Dep cases does not alter the $NH_3$ burden but leads to higher deposition and a shorter lifetime for both species.

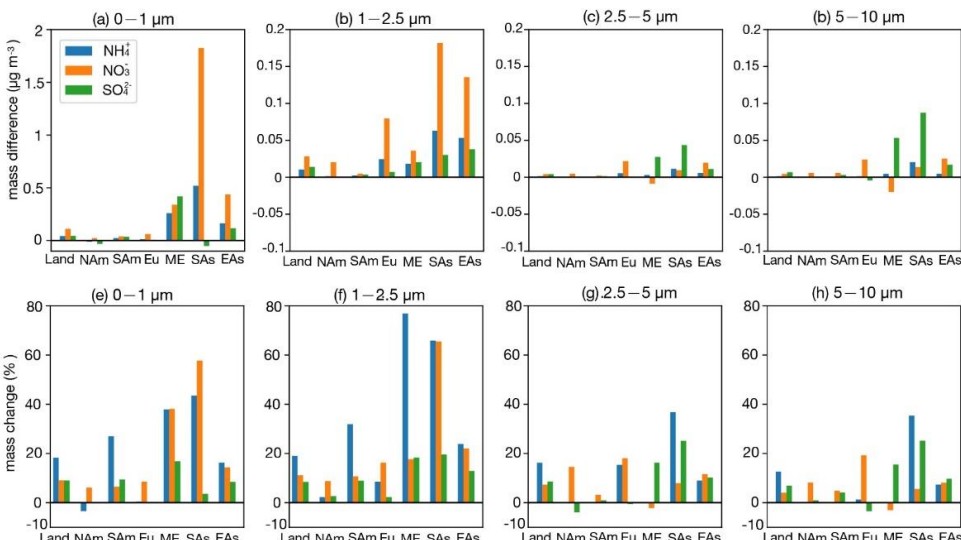

**Figure 7.** Bar plots for regional surface SNA mass concentration (µg m⁻³) absolute difference between CEDS case and base case in the four size ranges (a) – (d); change ratio in the four size ranges (e) – (h). The calculation of change ratio in the size range of 0–1 µm is based on the mask of 0.1 µg m⁻³, the change ratio in the size range of 1–2.5 µm, 2.5–5 µm and 5–10 µm is based on the mask of 0.05 µg m⁻³ (Land: global land; NAm: North America; SAm: South America; Eu: Europe; ME: Middle East; SAs: South Asia; EAs: East Asia).




Across land regions, a small increase in $NH_3$ emissions (CEDS case), along with rising $NO_x$ and $SO_2$ emissions, slightly raises $\varepsilon(NH_4^+)$ while marginally lowering $\varepsilon(NO_3^-)$ (Figure S15). The SNA mass concentration increases consistently across size ranges, with the most notable growth in $NH_4^+$ and $NO_3^-$ in the 1–2.5 µm range (19% and 11%, respectively) and $SO_4^{2-}$ in the submicron particles (9%). Under a larger $NH_3$ emission increase (Top-Dep case), $\varepsilon(NH_4^+)$ drops significantly, and $\varepsilon(NO_3^-)$ decreases slightly. The SNA response becomes more pronounced, with substantial increases in the 1–2.5 µm range ($NH_4^+$: 104%, $NO_3^-$: 41%, $SO_4^{2-}$: 23%), while $NO_3^-$ and $SO_4^{2-}$ decrease in coarser particles. These shifts are linked to higher $NH_3$ emissions in relatively clean, high-latitude regions with low $NH_3$ flux (Figure 1c), which elevate $NH_3$ concentrations (Figure S6) enhancing $NH_4^+$ formation (Figure S11).

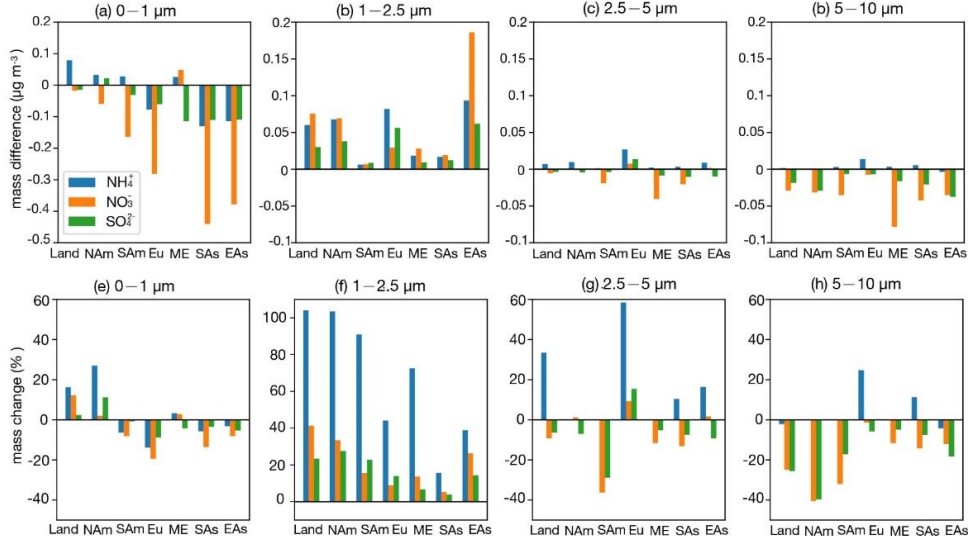

**Figure 8.** Bar plots for regional surface SNA mass concentration (µg m$^{-3}$) absolute difference between Top-Dep case and base case in the four size ranges (a) – (d); change ratio in the four size ranges (e) – (h). The calculation of change ratio in the size range of 0–1 µm is based on the mask of 0.1 µg m$^{-3}$, the change ratio in the size range of 1–2.5 µm, 2.5–5 µm and 5–10 µm is based on the mask of 0.05 µg m$^{-3}$ (Land: global land; NAm: North America; SAm: South America; Eu: Europe; ME: Middle East; SAs: South Asia; EAs: East Asia).

Regional analysis of SNA responses in Europe shows that a 23% reduction in $NH_3$ emissions, along with increases in $NO_x$ and $SO_2$ emissions (CEDS case), raises $\varepsilon(NH_4^+)$ slightly while causing a minor drop in $\varepsilon(NO_3^-)$. The SNA mass increases mainly in the 1–2.5 µm range, with a slight decrease in $SO_4^{2-}$ in coarser particles. Since Europe has abundant $NH_3$, reductions are offset by existing availability and rising $NO_x$ and $SO_2$ levels, leading to additional $NH_4NO_3$ and $(NH_4)_2SO_4$ formation. With higher $NH_3$ emissions (TopDep case), the response becomes more complex: SNA decreases in the submicron particles (Figure 8e) but increases in larger particles (Figure 8f-g), with the largest growth seen in $NH_4^+$ (~50%).

In East Asia, reducing $NH_3$ emissions while increasing $NO_x$ and $SO_2$ emissions (CEDS case) raises $\varepsilon(NH_4^+)$ and slightly increases $\varepsilon(NO_3^-)$. SNA grows across all particle sizes, with the most significant changes in 1–2.5 µm range (Figure 7f). A similar $NH_3$ reduction in the Top-Dep case produces comparable $\varepsilon(NH_4^+)$ and $\varepsilon(NO_3^-)$ changes, with



SNA mainly decreasing in the submicron particles. The response in East Asia resembles that of Europe, where
abundant $NH_3$ buffers SNA changes. In South Asia, $NH_3$ and $NO_x$ emissions rise in the CEDS case while $SO_2$
emissions decline, slightly lowering $\varepsilon(NH_4^+)$ and increasing $\varepsilon(NO_3^-)$. $NH_4^+$ and $NO_3^-$ concentrations grow across all
size ranges (Figure 8), with the largest $NH_4^+$ increase found in 1–2.5 µm particles (66%) and $NO_3^-$ rising in both
submicron and 1–2.5 µm particles (~60%). $SO_4^{2-}$ decreases in submicron particles but increases in coarser ones (25%).
Conversely, reducing $NH_3$ emissions in the Top-Dep case raises $\varepsilon(NH_4^+)$ and lowers $\varepsilon(NO_3^-)$, leading to SNA declines,
especially in submicron particles. South Asia, with abundant $NH_3$, shows $NO_3^-$ as the limiting factor for $NH_4NO_3$
formation, driving strong $\varepsilon(NO_3^-)$ and $NO_3^-$ responses in finer particles.

In North America, $NH_3$ and $SO_2$ emissions decrease, while $NO_x$ emissions slightly rise (CEDS case). $\varepsilon(NH_4^+)$
increases marginally, whereas $\varepsilon(NO_3^-)$ declines slightly. $NH_4^+$ and $SO_4^{2-}$ show minor reductions in submicron particles,
while $NO_3^-$ increases. In the Top-Dep case, a sharp rise in $NH_3$ emissions, mainly over Canada and Greenland,
significantly lowers $\varepsilon(NH_4^+)$, stabilizes $\varepsilon(NO_3^-)$, and increases $NH_4^+$ and $SO_4^{2-}$ in 1–2.5 µm particles. In South
America, a small $NH_3$ emission rise (CEDS case) has little effect on $\varepsilon(NH_4^+)$ or $\varepsilon(NO_3^-)$, resulting in minimal aerosol
composition changes. However, a substantial $NH_3$ increase (Top-Dep case) significantly boosts $NH_3$ concentrations,
reduces $\varepsilon(NH_4^+)$ and $\varepsilon(NO_3^-)$, and shifts aerosol partitioning to smaller particles, particularly in central regions.
Decreased $NO_3^-$ and $SO_4^{2-}$ in surrounding areas (Figures S12–13) suggest $NH_3$ is neutralizing transported species,
explaining the observed $\varepsilon(NO_3^-)$ reduction. Changes in other particle size ranges are minimal. In the Middle East, $NH_3$
emissions rise moderately (20%), along with slight $NO_x$ and $SO_2$ increases (CEDS case). $\varepsilon(NH_4^+)$ and $\varepsilon(NO_3^-)$ remain
stable, with $NH_4^+$ and $NO_3^-$ increasing mainly in submicron particles (~40%), while $SO_4^{2-}$ grows across sizes (~15%).
A minor $NH_3$ emission drop (Top-Dep case) reduces $NH_3$ concentrations but enhances SNA in 1–2.5 µm particles,
highlighting compensatory $NH_3$ effects in "$NH_3$ very rich" domains.

Overall, higher $NH_3$ emissions enhance SNA formation, particularly in the fine size ranges (0–1 µm and 1–2.5
µm). Greater $NH_4^+$ formation (e.g., Top-Dep case) depletes $NO_3^-$ and $SO_4^{2-}$ from coarse size ranges, leading to
decreases in 2.5–5 µm and 5–10 µm. In low-SNA regions (e.g., South America, Greenland), $NH_3$ increases have
limited SNA impacts but substantially elevate $NH_3$ concentrations. In "$NH_3$ very rich" regions (e.g., East Asia,
Europe), $NH_3$ reductions alone may still increase $\varepsilon(NH_4^+)$, promoting further SNA formation. These findings align
with Zou et al. (2024), emphasizing the greater effectiveness of $NO_x$ reductions in $PM_{2.5}$ mitigation compared to $NH_3$
or $SO_2$ control in ammonia-rich environments. Coordinated control of precursor emissions is crucial for mitigating air
pollution, especially in heavily polluted regions (Wen et al., 2024).

### 7.2 Aerosol acidity response

### 7.2.1 Size-resolved pH changes

Among atmospheric buffering agents (e.g., conjugate acid-base pairs like $NH_3/NH_4^+$ and $CO_2/HCO_3^-$, as well as
organic acids), the $NH_3/NH_4^+$ acid-base pair exhibits the largest buffering capacity for aerosols, dominating much of
the continental urban areas (Zheng et al., 2023; Zheng et al., 2020). When $NH_3$ emissions are entirely eliminated,
changes in oceanic pH values are negligible compared to the base case (noNH3 case, Table 5 and Figure S5). However,
aerosols over land become significantly more acidic, with pH values in the 0–1 µm size range dropping to -0.2 in



South Asia and -0.17 in the Middle East. The most substantial pH decreases occur in 0–1 µm aerosols, primarily in South Asia, East Asia, Europe, North America, and South America, consistent with the "NH$_3$-buffered regions" identified by Zheng et al. (2020).

Interestingly, while neither the Middle East nor northern Africa are categorized as "NH$_3$-buffered regions," noticeable pH decreases are observed in the 0–1 µm and 1–2.5 µm size ranges over the Middle East, including Egypt

and Libya. In these regions, the SNA is dominated by "SO$_4^{2-}$ rich" and "NO$_3^-$ rich" chemical domains (Figure 4), where NH$_4^+$ cannot fully neutralize the available NO$_3^-$ and SO$_4^{2-}$. This results in excess acidic components, particularly in the 0–1 µm and 1–2.5 µm size ranges (Figure S2). Without NH$_3$ emissions, the abundance of NO$_3^-$ and SO$_4^{2-}$ further increases aerosol acidity.

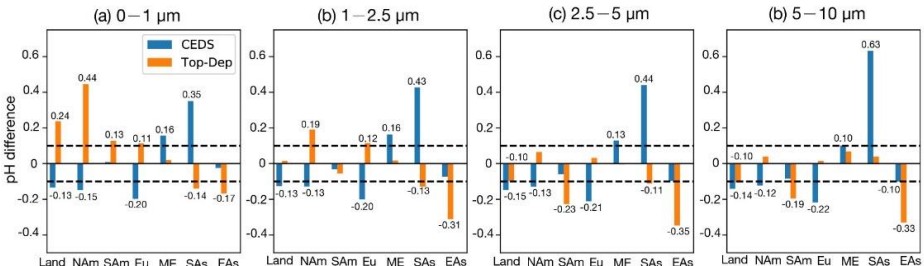


**Figure 9.** Bar plots for pH value absolute difference between CEDS /Top-Dep case and base case in the four size ranges (a) – (d), with the two dashed lines representing the value of 0.1 and -0.1, respectively (Land: global land; NAm: North America; SAm: South America; Eu: Europe; ME: Middle East; SAs: South Asia; EAs: East Asia).

Compared to the base case, size-resolved pH values in the CEDS case show an average decrease of -0.13 to -0.15 units over land. The most pronounced decreases occur in remote regions of the Northern Hemisphere (Figure S10), where NH$_3$ emission flux is relatively low. In contrast, the Top-Dep case exhibits an average pH increase of 0.24 units in the 0–1 µm range over land, driven by the higher NH$_3$ emission flux. This NH$_3$-driven alkalization effect is less pronounced in the 1–2.5 µm range due to counteracting acidification effects from newly formed acidic components,

as suggested by Zheng et al. (2024). Notably, pH decreases of -0.10 units are found in the 2.5–5 µm and 5–10 µm size ranges.

In East Asia, the pH changes in the CEDS case are concentrated in the 2.5–5 µm and 5–10 µm ranges (-0.10 units in both), with minimal changes in 0–1 µm and 1–2.5 µm. In the Top-Dep case, however, pH changes are more drastic across all size ranges: 0–1 µm (-0.17 units), 1–2.5 µm (-0.31 units), 2.5–5 µm (-0.35 units), and 5–10 µm (-0.33 units).

These discrepancies between cases highlight that a reduction in NH$_3$ emissions enhances aerosol acidity (as in the Top-Dep case), but this effect can be partially offset by shifts in SO$_4^{2-}$ and NO$_3^-$ mass concentrations (as in the CEDS case). For instance, in the CEDS case, decreases in SO$_4^{2-}$ and NO$_3^-$ concentrations result in pH rises of 0.15–0.25 units across all size ranges compared to the Top-Dep case.

Our simulation results align with other studies. For example, Song et al. (2019) found that a ~0.3 increase in

log$_{10}$[NH$_3$] contributed to a 0.3–0.4 unit rise in PM$_1$ pH in Beijing during winter between 2014/2015 and 2018/2019. Concurrent changes in aerosol composition (increased NO$_3^-$ and reduced SO$_4^{2-}$ and Cl$^-$) led to a modest 0.1 units pH



increase. Similarly, Zhou et al. (2022) observed a minor $PM_{2.5}$ pH decrease of -0.24 units in the Yangtze River Delta region from 2011 to 2019, despite significant changes in aerosol composition resulting from the Air Pollution Prevention and Control Action Plan.

**7.2.2 Drivers of pH Changes**

We further quantify the changes in pH values between the CEDS/Top-Dep cases and the base case, isolating the contributions from changes in $H^+$ and $H_2O$ concentrations (detailed in Section 2.2.2). The results are presented in Figures S16 and S17.

In the CEDS case, the decreased pH values across all four aerosol size ranges are primarily driven by changes in
$H^+$ concentrations, particularly over the remote regions of the Northern Hemisphere. The effects of $H_2O$ concentration changes are particularly relevant in the Middle East and South Asia. Notably, a substantial pH increase is observed in South Asia, with increases of 0.35, 0.43, 0.44, and 0.63 units in the 0–1 µm, 1–2.5 µm, 2.5–5 µm, and 5–10 µm size ranges, respectively. These increases are predominantly caused by changes in $H^+$ concentrations, which fully counteract the pH-decreasing effects of changes in $H_2O$ concentrations.

In the Top-Dep case, although changes in $H^+$ concentrations significantly enhance pH values in the 0–1 µm size range, the pH-decreasing effects of changes in $H_2O$ concentrations are distributed across all four size ranges over the entire land surface. In East Asia, both $H^+$ and $H_2O$ concentration changes contribute substantially to the observed pH decreases, highlighting their combined impact.

In summary, $NH_3$ emissions play a crucial role in maintaining terrestrial aerosols at moderately acidic levels,
particularly in the fine-mode size range of 0–1 µm. When $NH_3$ emissions are completely removed in the model, the largest pH decreases are found in this size range. Our findings underscore that, in the fine-mode ranges (0–1 µm and 1–2.5 µm), pH changes closely correspond to variations in $NH_3$ emissions, reflecting the distinct sensitivity of size-resolved pH to $NH_3$ levels. In the coarser size ranges (2.5–5 µm and 5–10 µm), however, pH changes are governed by the competing influences of $H^+$ and $H_2O$ concentration changes.

**8. Conclusion**

This study applied three distinct ammonia ($NH_3$) emission schemes to the global atmospheric chemistry and climate model (EMAC) to assess the impact of $NH_3$ emissions on the size-resolved sulfate-nitrate-ammonium (SNA) aerosol composition and aerosol acidity. The emission schemes included two bottom-up inventories, and one inventory updated through a top-down approach. By simulating the size ranges of 0–1 µm, 1–2.5 µm, 2.5–5 µm, and 5–10 µm,
the study provides a comprehensive analysis of the role of $NH_3$ emissions in influencing global aerosol composition and acidity.

The model accurately captures the distribution of global $NH_3$ hotspots, but comparisons with observational datasets reveal positive biases in $NH_3$ concentrations and underestimations of $NH_4^+$ in regions such as China, North America, and Europe. This discrepancy suggests inefficiencies or oversimplifications in the $NH_3/NH_4^+$ partitioning treatment,
with insufficient $NH_4^+$ generated despite $NH_3$ availability. In East and Southeast Asia, $NH_4^+$ concentrations are well-simulated, but significant discrepancies were found for $NO_3^-$ and $SO_4^{2-}$, likely due to the absence of heterogeneous



oxidation processes in the model. The simulated global $NH_3$ burden and lifetime are higher than reported in related studies, attributed to the wet deposition scheme used in EMAC which accounts for pH adjustments for $NH_3$ dissolution.

Over oceans, $NH_3$ is entirely converted to $NH_4^+$, while $SO_4^{2-}$ remains partially neutralized due to low $NH_3$ emissions and high $SO_4^{2-}$ levels from biogenic dimethyl sulfide (DMS) emissions and $NO_x$ and $SO_2$ from shipping. Marine $SO_4^{2-}$ and $NO_3^-$ are dominant in the super-micron modes, with mass fractions of 70% and 58%, respectively. On land, $SO_4^{2-}$ is generally fully neutralized by $NH_3$, except in regions such as northern Russia, central Africa, and the Arabian Peninsula. Terrestrial $NO_3^-$ is also largely neutralized by $NH_3$, resulting in over 60% of SNA concentrated in

the sub-micron mode. In polluted areas such as East and South Asia, sub-micron SNA fractions exceed 70%, with South Asia exhibiting nearly 90% sub-micron $NH_4^+$. In contrast, the Middle East, dominated by dust and with minor $NH_3$ emissions, shows incomplete neutralization of $SO_4^{2-}$ and $NO_3^-$. Here, nearly 70% of $NO_3^-$ resides in the super-micron modes, while $NH_4^+$ and $SO_4^{2-}$ dominate the sub-micron mode with respective mass fractions of 96% and 79%.

     In the Northern Hemisphere, terrestrial aerosols are generally more acidic than marine aerosols, except in desert

regions. Remote marine and desert aerosols remain neutral due to the alkaline influence of sea salt and non-volatile cations in dust, which enhance water uptake and neutralize $SO_4^{2-}$ and $NO_3^-$. However, at high latitudes, marine aerosols become more acidic due to the long-range transport of anthropogenic pollutants from continental sources. The 0–1 µm size range exhibits higher pH values than the 1–2.5 µm range in many regions, a trend influenced by several factors. These include the assumption of a thermodynamically stable aerosol phase in the ISORROPIA model,

the high sensitivity of aerosol acidity to $NH_3$ in the 0–1 µm range and observed lower concentrations of acidic components ($SO_4^{2-}$ and $NO_3^-$) in this size range compared to larger sizes.

     An 18% increase in $NH_3$ emissions over land leads to a significant increase in SNA concentrations in the 1-2.5 µm size range ($NH_4^+$: 104%, $NO_3^-$: 41%, $SO_4^{2-}$: 23%), while coarse mode SNA (2.5-10 µm) decreases. However, changes in size-resolved pH remain minimal, with the largest increase of 0.24 pH units occurring in the 0-1 µm range. In

regions with low SNA, such as South America and Greenland, increased $NH_3$ emissions only lead to higher $NH_3$ concentrations due to the limited availability of $HNO_3$ and $H_2SO_4$ for SNA formation. Conversely, in $NH_3$-rich regions such as East Asia and Europe, reductions in $NH_3$ emissions trigger compensatory effects, as excess $NH_3$ and available $SO_2$ and $NO_x$ help to maintain SNA formation despite the reduced $NH_3$ supply. The buffering effect of $NH_3$ plays a crucial role in stabilizing aerosol acidity in densely populated areas, mitigating the effects of fluctuations in precursor

gases.

     In summary, this study underscores the critical influence of $NH_3$ emissions on global aerosol composition and acidity, with pronounced impacts on the size-resolved composition and properties of SNA aerosols. $NH_3$ emissions significantly modulate aerosol acidity, particularly in sub-micron ranges, highlighting the sensitivity of fine-mode aerosols to $NH_3$ levels. By buffering changes in aerosol pH, $NH_3$ emissions contribute to maintaining a relatively

stable aerosol acidity, especially in heavily populated and industrialized regions. These findings emphasize the importance of accurately representing $NH_3$ dynamics in models for predicting atmospheric chemistry and climate interactions.



## Code and Data Availability

The usage of MESSy (Modular Earth Submodel System) and access to the source code is licensed to all affiliates of institutions which are members of the MESSy Consortium. Institutions can become a member of the MESSy Consortium by signing the "MESSy Memorandum of Understanding". More information can be found on the MESSy Consortium website: http://www.messy-interface.org (last access: 31 January 2025). The code used in this study has been based on MESSy version 2.55 and is archived with a restricted access DOI (https://doi.org/10.5281/zenodo.8379120, The MESSy Consortium, 2023). The data produced in the study is available from the authors upon request.

## Acknowledgements

The work described in this paper has received funding from the Initiative and Networking Fund of the Helmholtz Association through the project "Advanced Earth System Modelling Capacity (ESM)". The authors gratefully acknowledge the Earth System Modelling Project (ESM) for funding this work by providing computing time on the ESM partition of the supercomputer JUWELS (Alvarez, 2021) at the Jülich Supercomputing Centre (JSC). Xurong Wang has been supported by the China Scholarship Council (CSC).

## Competing Interests

At least one of the (co-)authors is a member of the editorial board of Atmospheric Chemistry and Physics.

## Author Contributions

XW and VAK planned the research and wrote the manuscript. XW performed the simulations and analyzed the results, assisted by VAK and APT. XW and APT collected the observational data and conducted the model evaluation. XW and ZL designed the Top-Dep case. All the authors discussed the results and contributed to the paper.

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
