# Peer review of "The influence of ammonia emission inventories on the sizeresolved global atmospheric aerosol composition and acidity"

_EGUsphere, 2025_

## Referee Comment (RC2)

**Comments for "egusphere-2025-527: The influence of ammonia emissions on the size-resolved global atmospheric aerosol composition and acidity" by Wang et al. (2025)**

Wang et al. presents a comprehensive investigation of how ammonia ($NH_3$) emissions affect size-resolved aerosol composition and acidity on a global scale. Using the EMAC atmospheric chemistry-climate model with three different ammonia emission schemes, the authors analyze the complex interactions between ammonium, sulfate and nitrate in different sizes, geographic regions, and chemical environments. Research advances our understanding of atmospheric aerosol dynamics and has significant implications for air quality management and climate modeling. I recommend publication after addressing the following comments.

**Research questions and modeling approach lack clarity**

The Introduction provides a comprehensive overview of aerosol emission trends and how PM2.5 components respond to different clean air policies. It also notes that these responses vary depending on particle size ranges. However, the progress in modeling this size-resolved response is not clearly presented. The intended model for use in this work and its suitability are not well explained, and the scientific question lacks clarity. The authors are encouraged to clearly articulate the rationale for selecting the model, i.e., why EMAC is appropriate for this question?

**The role of organic aerosols in affecting aerosol pH**

Although the main goal of this work is to study the size-resolved SNA and pH response to different ammonium emission inventories, it would be beneficial to include some discussions on the role of organics in influencing these outcomes. As significant components of aerosol particles with diverse hygroscopic properties, organic aerosols can absorb water and impact both aerosol liquid water content and pH. Including a discussion on how organics might alter the size-resolved response would strengthen the analysis.

For example, the reported 104% increase in $NH_4^+$ in response to an 18% rise in ammonia emissions could not solely from interactions with sulfate and nitrate, but may partly result from reactions between ammonia and organic acids (e.g., forming ammonium oxalate). These processes can influence pH, especially in the 0–1 um range. Neglecting the role of organics risks overattributing observed effects to SNA alone.

**Minor comments:**

Line 170: The cases are not clearly defined in the texts that describe Table 1. What are noNH3 and Meta cases? You can briefly introduce why you conduct these two cases here. Is Top-Dep case using the Top-down scheme?

Line 207: The symbols and Italic fonts used in the texts and equations throughout the paper, such as $E_{NH_3,mod}$ do not follow standard scientific writing conventions. For guidance, you may refer to this document.:https://iupac.org/wp-content/uploads/2016/01/ICTNS-On-the-use-of-italic-and-roman-fonts-for-symbols-in-scientific-text.pdf

Line 217: Remove the dot after number 74.

Line 230: Using lighter background colors in Figure 1(a) would improve clarity and make the hotspots easier to distinguish.

Line 235 and 245: Since these are comparative descriptions rather than time series trends, I'll avoid using "increase" and instead opt for terms like "overestimate" or "biases".

Line 295: references for IPCC(2023)?

Line 351: Typos in this paragraph. Change $SO_4^-$ to $SO_4^{2-}$.

line 505: References for the statement:"high-latitude marine aerosols are more acidic ..."?

Line 538: Table 9?

Line 539: It would be better to provide more context for the motivation of conducting the noNH3 case earlier in the text-when introducing the cases in Table 1—rather than introducing it abruptly here.

Line 556: Since the effects of different ammonia emission scheme are a crucial aspect of this research, and the title is "The Influence of Ammonia Emissions...," it would be more appropriate to move the sentences discussing the importance of the ammonia emission inventory and its effects earlier.

Line 605: The figure captions for Figure 7 and Figure 8 are almost the same. You can change the caption of figure 8 to "The same as figure 7, but for the difference between Top-Dep case and base case. "

---

## Author Comment (AC1)

**Authors' response to comments made by anonymous reviewer #1:**

*This study examines the role of NH₃ emissions in global aerosol chemical composition and acidity. The study uses the EMAC model with three different emissions scenarios to quantify effects on size-resolved inorganic aerosol composition and pH. The study observes a link between NH₃ emissions and pH, with complex effects that vary regionally. The study is the first, to my knowledge, to examine predictions of aerosol pH and the sensitivity of pH to different emissions scenarios on a global scale. The study highlights some current model limitations that contribute to challenges predicting pH. The manuscript is very well written and nicely organized. The scope is certainly a fit for ACP and I believe it will be of interest to a broad audience once a number of issues are addressed.*

We thank the reviewer for the thoughtful and positive comments on our manuscript. We greatly appreciate the constructive feedback provided, which has helped us improve the clarity and overall quality of the manuscript. Below, we provide a point-by-point response (in black) to all the issues raised (in red).

**Specific Comments:**

*1.     A significant issue with the manuscript is the predictions of pH in arid regions of the globe. Thermodynamic equilibrium models, including ISORROPIA, are challenged to represent aerosol pH when relative humidity is low (e.g., see the extensive discussion in Pye et al. (2020)). The manuscript contains significant discussion of results in dry, arid regions – e.g., the Middle East, and over desert regions. The physical interpretation of these results is ambiguous unless the predictions of ALWC and aerosol pH in these regions are more closely scrutinized. The associated discussions likely need substantial revision, or at least more discussion about the potential problems of such predictions under low RH conditions.*

We thank the reviewer for highlighting the limitations of thermodynamic equilibrium models under low relative humidity (RH) conditions. We fully acknowledge that predicting aerosol pH in arid regions remains a significant challenge, as discussed in Pye et al. (2020). Discrepancies among thermodynamic models tend to grow as RH decreases, primarily due to differences in the assumptions about activity coefficients. For instance, ISORROPIA assumes constant mean activity coefficients and a unity activity coefficient for $H^+$, which results in relatively invariant and often lower pH predictions under dry conditions. In contrast, models such as E-AIM incorporate RH-dependent variations in single-ion activity coefficients for both solutes and solvents, potentially leading to more dynamic pH behavior as RH changes. Additionally, the choice between stable and metastable phase state assumptions introduces further uncertainty. Under low RH (e.g., <35%), the metastable assumption allows for supersaturated solutions, enabling pH calculations even at low aerosol liquid water content (ALWC), often resulting in low pH values. Conversely, the stable state permits salt crystallization, and in cases where the aerosol becomes fully solid, pH may no longer be defined. To assess the sensitivity of our results to this assumption, we conducted a simulation using the metastable assumption (Meta case; see Table 1), with results presented in Table 9 and Figure 7. We have revised the relevant discussion sections (especially section 2.2.1) to better reflect these uncertainties and to avoid overinterpreting  pH predictions in arid regions.

*2.    The title and the abstract are misleading because the different emission scenarios vary $SO_2$ and $NO_x$ emissions at the same time as $NH_3$. Therefore, interpreting the results is more complex than a typical modeling sensitivity analysis where one factor is varied while all other factors are held constant. In the case of pH, it is not straightforward to attribute the observed changes to the differences in $NH_3$ because the precursors for aerosol sulfate and nitrate also changed simultaneously.  With the emissions in $SO_2$, $NO_x$, and $NH_3$ changing in different directions (some regions, these go up, in other regions, they go down, and not always together), the results were quite complex and not easy to interpret. For example, a conclusion of the study succinctly stated in Line 707 is: "pH changes closely correspond to variations in $NH_3$ emissions", however, the study is not really able to derive the quantitative relationship in each region because of the concurrent changes in $SO_2$ and $NO_x$. Ultimately, I think the authors need to do more to facilitate interpretation of the results and isolate the effects of individual species on the observed changes in pH, though this is not easy.*

We sincerely thank the reviewer for this insightful comment. We acknowledge that the emission scenarios used in our study involve simultaneous changes in $SO_2$ and $NO_x$ emissions, while $NH_3$ emissions are not directly varied but are instead represented using three different emission inventories: two bottom-up inventories (CAMS and CEDS_GBD) and one top-down inventory (Luo et al., 2022). This approach allows us to explore the sensitivity of aerosol pH to plausible variations in $NH_3$ emissions as reflected by different inventory methodologies, rather than through isolated perturbation experiments. We agree that the co-variation of $SO_2$, $NO_x$, and $NH_3$ across these inventories adds complexity to the interpretation of pH changes, and we have revised the manuscript to clearly articulate this point. The title has been revised to "*The influence of ammonia emission inventories on the size-resolved global atmospheric aerosol composition and acidity*" to more accurately reflect the study. Furthermore, we have expanded the discussion in Section 7 to more explicitly address the interactions among $NH_3$, $SO_2$, and $NO_x$, and we have revised the abstract and conclusion to better reflect the complexity of interpreting pH responses under varying emission inputs and avoid over-attributing pH changes to $NH_3$ alone.

*3.    The authors acknowledge that the size-resolved pH predictions do not follow the expected trend in many cases, as pH does not systematically decrease with decreasing particle diameter for the different size bins in many locations.  Only when $NH_3$ emissions are eliminated does the model predict more acidic smaller particles. The model errors in predictions of $NH_3/NH_4^+$ partitioning suggest there are associated errors in the predictions of aerosol pH. Much more discussion of this point is warranted.*

Indeed, our model does not consistently reproduce the expected trend of decreasing pH with decreasing particle size across all regions. This trend is only clearly observed in the no-$NH_3$ scenario, where smaller particles are predicted to be more acidic than those in the coarse mode. To further investigate this issue, we compared the simulated SNA composition and $NH_3/NH_4^+$ partitioning ratios with available observational data. This comparison revealed discrepancies that can contribute to positive biases in pH predictions of submicron particles, particularly an overestimation of $NH_3$ partitioning into the aerosol phase and an underestimation of acidic components such as sulfate and nitrate, especially over Europe. We have incorporated a more detailed discussion of these findings in Section 6, emphasizing the implications of partitioning errors on size-resolved pH predictions. We highlight the need for caution when interpreting pH

trends in regions with high ammonium content in fine aerosol modes, where model uncertainties in gas-particle partitioning can significantly influence the predicted acidity.

*4.  A comparison of stable and metastable mode results from thermodynamic models has been done before. However, it has never been done for global simulations, so the present results are quite important because of their scale. I encourage the authors to expand on this discussion and to consider moving Fig. S4 to the main manuscript.*

We thank the reviewer for this valuable suggestion. Karydis et al. (2021) conducted a global-scale comparison of stable and metastable state assumptions and found that the stable-state assumption leads to global average pH values approximately 0.5 units higher than those under metastable conditions. Our study builds on this by providing additional insights into size-resolved pH differences and their regional variability. Our results corroborate the global-scale findings of Karydis et al. (2021) but also reveal that discrepancies between stable and metastable assumptions can be substantially larger, exceeding 2 pH units, in regions with high concentrations of crustal cations and persistently low RH, such as South Asia and the Middle East. Moreover, our analysis highlights how these differences vary across aerosol size modes, offering a more detailed understanding of the thermodynamic behavior of aerosols under varying environmental conditions. In response to the reviewer's recommendation, we have moved Figure S4 to the main manuscript as Figure 7, and we have expanded the discussion in Section 6 to more thoroughly examine the implications of phase state assumptions for interpreting aerosol acidity, particularly in arid and dust-influenced regions.

*5.  Section 2.2.2: it is really not accurate to frame the discussion around $H^+$ and $H_2O$, only. Other particle components can affect the aerosol pH by affecting the $H^+$ activity coefficient. Although ISORROPIA assumes an $H^+$ activity coefficient of unity, other models that solve for $\gamma_{H+}$ would have an effect on pH from other aerosol components.*

We agree that it is not fully accurate to frame the discussion of aerosol pH solely in terms of $H^+$ and $H_2O$. As correctly noted, other aerosol components, such as sulfate, nitrate, and organics, can significantly influence the ionic strength of the aerosol solution and, consequently, the activity coefficient of $H^+$. While ISORROPIA assumes a constant $H^+$ activity coefficient of unity, this simplification limits its ability to capture composition-dependent effects on pH, particularly under low relative humidity (RH) conditions where ionic strength can vary substantially. In our study, following ISORROPIA's framework, pH is approximated based on free-$H^+$ molality under the assumption of $\gamma_H^+ = 1$. Accordingly, our discussion in Section 2.2.2 focuses on variations in free-$H^+$ concentration as the primary driver of pH changes. We have revised Section 2.2.1 to explicitly acknowledge this limitation and to clarify that our pH estimates do not account for composition-dependent variations in activity coefficients. We also highlight that this simplification may contribute to discrepancies when comparing with models that include a more detailed thermodynamic treatment.

*6.    Table 7: why is the SE-USA case from Pye et al. (2020) not included in this comparison? Aerosol pH in this region has been studied extensively and should provide a good point of comparison for the present study.*

We appreciate the reviewer's suggestion to include the SE-USA case from Pye et al. (2020) as a point of comparison. We agree that this region has been extensively studied and offers valuable insights into the aerosol pH behavior. However, the observational period in Pye et al. (2020) (June 6 – July 14, 2013) does not overlap with our simulation period (2009–2012), which limits the direct comparability of the datasets. Nevertheless, we now refer to the findings from that study in section 4.2 to provide additional context and to support the interpretation of our results in the southeastern U.S. region.

*7.    For the comparison to the field-derived pH in Xi'an shown in Table 7, the authors are encouraged to consult Guo et al. (2017), who provide a different estimate of aerosol pH in Xi'an. It is not reasonable to request the present manuscript to arbitrate this disagreement, however, the authors should be aware of different pH estimates for this region.*

Similar to our response to Comment 6, the observational period in Guo et al. (2017), which focuses on winter 2013, falls outside our simulation period (2009–2012). Nonetheless, we recognize the importance of acknowledging alternative pH estimates for Xi'an. To reflect this, we have added a reference to Guo et al. (2017) in section 4.2, noting the differences in observational periods and highlighting the variability in reported pH values for this region. While our study does not attempt to reconcile these differences, we agree it is important to be aware of them when interpreting model–observation comparisons.

*8.    Overall comment: reconsider the number of sig figs used in many cases. E.g., in Section 5.3 reporting $NH_3$ emissions to 0.01 Tg and reporting $NH_3$ lifetimes to the 0.01 day do not likely reflect uncertainties in these values.*

We agree with the reviewer's suggestion regarding numerical precision. We have revised the number of significant figures in Tables 2, 8, and S1, and throughout the related text. Emissions and deposition are now reported to 0.1 Tg $yr^{-1}$, burdens to 0.1 Tg, and lifetimes to 0.1 days.

*9.    Lines 72 – 74: this sentence needs revision – what does "excess $NH_3$ released to the atmosphere" really mean.*

Due to reductions in $SO_2$ and $NO_x$ emissions, the atmospheric formation of $H_2SO_4$ and $HNO_3$ has declined. Consequently, less $NH_3$ is required to neutralize these acids. At the same time, $NH_3$ emissions have remained stable or slightly increased, resulting in a relative surplus of $NH_3$ in the atmosphere. We have revised the sentence to clarify this point.

*10.    Color scale of Fig. 1 was quite difficult to determine the magnitude of the changes in many regions.*

We have reconfigured the data intervals and the corresponding color bar to enhance visual clarity and improve the distinction between different regional patterns and gradients.

*11.   Line 411 and 430 (and elsewhere): best not to use phrases like this…global pH values show that sulfuric acid is rarely fully neutralized. See also Guo et al. (2017).*

We have revised this terminology throughout the manuscript to use more appropriate language.

*12.   Paragraph lines 549 – 555: I do not follow the discussion in this paragraph.*

We believe that part of the discrepancy in size-resolved pH calculations between models may arise from differences in how particle size distributions and gas-particle partitioning are treated. For example, the EMAC model uses a lognormal size distribution and applies ISORROPIA separately to each size mode to calculate gas-aerosol partitioning. In contrast, other studies (e.g., Kakavas et al., 2021) employ sectional approaches, where gas-aerosol partitioning is first performed on the bulk aerosol phase, and the resulting condensed mass is then distributed across size bins based on the available surface area. These fundamental differences in modeling assumptions can lead to variations in the predicted distribution of aerosol components across size ranges, which in turn affects the calculated size-resolved pH. To clarify this point, we have revised the text to better explain how differences in size distribution and partitioning methods may contribute to the observed discrepancies.

*13.   Line 562: the effect of NH$_3$ emissions on SNA formation has been studied for decades. Also, I would not categorize the effect of NH$_3$ emissions on pH as the "subject of debate," but rather understudied.*

Thank you for the helpful suggestion. We agree that the role of NH$_3$ emissions in SNA formation has been extensively studied over the past decades. Additionally, we acknowledge that describing the effect of NH$_3$ on aerosol pH as a "subject of debate" may be misleading. A more accurate characterization is that this topic remains understudied, particularly in terms of its size-resolved and region-specific impacts in global modeling frameworks. Accordingly, we have revised the sentence in revised manuscript to better reflect the current state of knowledge.

*14.   Line 704: this is not true in terrestrial regions where highly acidic (e.g., pH < 2) particles are observed or predicted.*

We apologize for the confusion. The original statement was intended to compare high-latitude marine aerosols to those over remote ocean regions, not to terrestrial regions. We have revised the text to clarify that high-latitude marine aerosols are more acidic compared to aerosols over remote oceanic regions.

**Technical Corrections:**

*1.  Line 67: trend should be plural.*

Thank you for pointing this out. We have corrected the typo.

*2. Line 133: should 'of' be added after 'number'?*

Thank you for the suggestion. We have added "of" after "number" to improve clarity.

*3. Table 2 header: 'cases' is repeated.*

We have removed the repeated word.

*4. Line 653: do the authors mean 'marine aerosol' instead of 'oceanic'?*

Yes, "marine aerosol" is more accurate in this context. We have revised the text accordingly.

**References**

Guo, H., Weber, R. J., and Nenes, A.: High levels of ammonia do not raise fine particle pH sufficiently to yield nitrogen oxide-dominated sulfate production, Scientific Reports, 7, 12109, 10.1038/s41598-017-11704-0, 2017a.

Pye, H. O. T., Nenes, A., Alexander, B., Ault, A. P., Barth, M. C., Clegg, S. L., Collett Jr, J. L., Fahey, K. M., Hennigan, C. J., Herrmann, H., Kanakidou, M., Kelly, J. T., Ku, I. T., McNeill, V. F., Riemer, N., Schaefer, T., Shi, G., Tilgner, A., Walker, J. T., Wang, T., Weber, R., Xing, J., Zaveri, R. A., and Zuend, A.: The acidity of atmospheric particles and clouds, Atmos. Chem. Phys., 20, 4809-4888, 10.5194/acp-20-4809-2020, 2020.

---

## Author Comment (AC2)

**Authors' response to comments made by anonymous reviewer #2:**

**Summary**

*Wang et al. presents a comprehensive investigation of how ammonia (NH3) emissions affect size-resolved aerosol composition and acidity on a global scale. Using the EMAC atmospheric chemistry-climate model with three different ammonia emission schemes, the authors analyze the complex interactions between ammonium, sulfate and nitrate in different sizes, geographic regions, and chemical environments. Research advances our understanding of atmospheric aerosol dynamics and has significant implications for air quality management and climate modeling. I recommend publication after addressing the following comments.*

We thank the reviewer for the thoughtful and positive comments on our manuscript. We appreciate the constructive feedback provided which helped us improve the clarity and quality of the manuscript. Below is a point-by-point response (in black) to all the points raised (in red).

**Specific Comments:**

**1. Research questions and modeling approach lack clarity**

*The Introduction provides a comprehensive overview of aerosol emission trends and how $PM_{2.5}$ components respond to different clean air policies. It also notes that these responses vary depending on particle size ranges. However, the progress in modeling this size-resolved response is not clearly presented. The intended model for use in this work and its suitability are not well explained, and the scientific question lacks clarity. The authors are encouraged to clearly articulate the rationale for selecting the model, i.e., why EMAC is appropriate for this question?*

We thank the reviewer for this valuable feedback. We recognize the importance of clearly articulating both the scientific motivation and the rationale behind our modeling approach. In response, we have thoroughly revised the manuscript to clarify the research objectives and the suitability of the EMAC model for addressing our questions. Specifically, we have updated the final paragraph of the Introduction to clearly state the scientific questions driving this study, including the size-resolved response of aerosol composition and acidity to changes in precursor emissions. Furthermore, we justify the selection of the EMAC model in the introduction and model description (i.e., section 2), emphasizing its state-of-the-art capability to simulate global-scale aerosol–chemistry–climate interactions with size resolution, and its integration with the ISORROPIA-II thermodynamic module for aerosol pH estimation. ISORROPIA-II is a widely used thermodynamic model well-suited for simulating aerosol pH, as it efficiently handles size-resolved inorganic aerosol systems under varying humidity conditions. While it simplifies some aspects (e.g., assuming unity for certain activity coefficients), it has been extensively validated and remains a practical choice for large-scale pH simulations. These revisions aim to provide a more coherent narrative linking the research context, modeling framework, and study objectives.

**2. *The role of organic aerosols in affecting aerosol pH**

*Although the main goal of this work is to study the size-resolved SNA and pH response to different ammonium emission inventories, it would be beneficial to include some discussions on the role of organics in influencing these outcomes. As significant components of aerosol particles with diverse hygroscopic properties, organic aerosols can absorb water and impact both aerosol liquid water content and pH. Including a discussion on how organics might alter the size-resolved response would strengthen the analysis.*

*For example, the reported 104% increase in $NH_4^+$ in response to an 18% rise in ammonia emissions could not solely from interactions with sulfate and nitrate, but may partly result from reactions between ammonia and organic acids (e.g., forming ammonium oxalate). These processes can influence pH, especially in the 0–1 μm range. Neglecting the role of organics risks overattributing observed effects to SNA alone.*

We appreciate the reviewer's suggestion to further discuss the role of organic aerosols in influencing aerosol pH. While the primary focus of this study is on the size-resolved response of SNA aerosols to changes in ammonia emissions, we agree that organics can also play a role, particularly through their contribution to aerosol liquid water content (ALWC) and, to a lesser extent, their influence on hydrogen ion activity. In our model, the effect of water-soluble organic aerosols on ALWC is accounted for via the GMXe module, which includes both inorganic and organic contributions. Organic aerosol formation is simulated using the ORACLE module, and the associated water uptake is calculated assuming a κ-hygroscopicity value of 0.14 for all organic components (Tsimpidi et al., 2014). This influences the total aerosol water content used in pH calculations. However, our model in the present set-up does not account for chemical interactions between ammonia and organic acids (e.g., formation of ammonium oxalate), and it treats the inorganic and organic aerosol phases independently. Consequently, while organics can indirectly affect pH through water uptake, changes in $NH_3$ emissions do not influence organic aerosol formation or the associated water content in our simulations. Therefore, the reported increase in $NH_4^+$ is attributed solely to interactions with inorganic aerosol components. Although the effect of organics on hydrogen ion activity coefficients is not explicitly included, previous studies have shown that water-soluble organic aerosols exert only a minor influence on aerosol pH. For example, Pye et al. (2018) estimated that organic-associated hydrogen ions increase $PM_{2.5}$ pH by only ~0.1 units, while Vasilakos et al. (2018) found that organics induce pH deviations of less than 2% across a range of compounds and environmental conditions. These findings are consistent with other studies (Battaglia Jr et al., 2019; Pye et al., 2020; Guo et al., 2015; Liu et al., 2017), supporting the limited role of organics in modulating aerosol acidity. We have added this clarification to Section 2.2.1, along with a discussion of the model's limitations in representing ammonia–organic interactions.

**Minor comments:**

1. *Line 170: The cases are not clearly defined in the texts that describe Table 1. What are noNH$_3$ and Meta cases? You can briefly introduce why you conduct these two cases here. Is Top-Dep case using the Top-down scheme?*

Thank you for pointing this out. In the noNH$_3$ case, all ammonia emissions are turned off. The Meta case is identical to the base case, except that the ISORROPIA model is run in metastable mode. The Top-Dep

case applies the top-down emission scheme. We have added a brief explanation of these simulation cases in lines 176–180 to clarify their purpose and setup.

2. *Line 207: The symbols and Italic fonts used in the texts and equations throughout the paper, such as $E_{NH3,mod}$ do not follow standard scientific writing conventions. For guidance, you may refer to this document.:https://iupac.org/wpcontent/uploads/2016/01/ICTNS-On-the-use-of-italic-and-roman-fonts-for-symbols-in-scientific-text.pdf*

We sincerely appreciate the reviewer's guidance on the proper use of italic and roman fonts for scientific symbols. Following the IUPAC recommendations, we have systematically revised the formatting of all symbols throughout the text and equations to ensure consistency with standard conventions.

3. *Line 217: Remove the dot after number 74.*

Done.

4. *Line 230: Using lighter background colors in Figure 1(a) would improve clarity and make the hotspots easier to distinguish.*

Thank you for the suggestion. We have redrawn Figure 1(a) using a lighter background color to enhance visual clarity and improve the distinction of hotspot regions.

5. *Line 235 and 245: Since these are comparative descriptions rather than time series trends, I'll avoid using "increase" and instead opt for terms like "overestimate" or "biases".*

Thank you for the helpful suggestion. We have revised the wording accordingly.

6. *Line 295: references for IPCC(2023)?*

We have added the appropriate reference for IPCC in the reference list.

7. *Line 351: Typos in this paragraph. Change $SO_4^-$ to $SO_4^{2-}$ .*

Corrected.

8. *line 505: References for the statement: "high-latitude marine aerosols are more acidic ..."?*

High-latitude marine aerosols are generally more acidic than those over remote ocean regions, primarily due to the long-range transport of anthropogenic pollutants such as $H_2SO_4$ and $HNO_3$ from continental sources. This enhanced acidity has been observed and simulated in several studies. For example, Karydis

et al. (2021) conducted a global modeling study showing that aerosol pH in high-latitude marine regions is significantly influenced by anthropogenic outflow, particularly from Europe and North America. Similarly, Myhre et al. (2013) discuss how anthropogenic aerosols, including sulfate and nitrate, are transported to remote marine environments, altering their chemical composition and acidity. We have updated the text in the revised manuscript to include these references.

9. *Line 538: Table 9?*

We have corrected the table number accordingly.

10. *Line 539: It would be better to provide more context for the motivation of conducting the noNH₃ case earlier in the text-when introducing the cases in Table 1—rather than introducing it abruptly here.*

Thank you for the suggestion. We have added a brief explanation of the noNH₃ case in lines 176–180 to improve the flow of the manuscript.

11. *Line 556: Since the effects of different ammonia emission scheme are a crucial aspect of this research, and the title is "The Influence of Ammonia Emissions...," it would be more appropriate to move the sentences discussing the importance of the ammonia emission inventory and its effects earlier.*

We appreciate this thoughtful suggestion. After careful consideration, we have decided to retain the current structure to preserve the logical flow of the manuscript. Our approach begins with the model and observational datasets (Sections 2 and 3), followed by evaluation of the base case (Section 4), then analysis of global and regional aerosol chemical regimes (Sections 5 and 6), and finally the emission sensitivity analysis (Section 7). Reordering Section 7 earlier could disrupt this flow, as the interpretation of the emission scenario results depends on understanding the performance and limitations of the base case.

12. *Line 605: The figure captions for Figure 7 and Figure 8 are almost the same. You can change the caption of figure 8 to "The same as figure 7, but for the difference between Top-Dep case and base case.*

Thank you for the suggestion. We have revised the caption for Figure 8 accordingly.

**References**

Battaglia Jr, M. A., Weber, R. J., Nenes, A., and Hennigan, C. J.: Effects of water-soluble organic carbon on aerosol pH, Atmos. Chem. Phys., 19, 14607-14620, 10.5194/acp-19-14607-2019, 2019.

Guo, H., Xu, L., Bougiatioti, A., Cerully, K. M., Capps, S. L., Hite Jr, J. R., Carlton, A. G., Lee, S. H., Bergin, M. H., Ng, N. L., Nenes, A., and Weber, R. J.: Fine-particle water and pH in the southeastern United States, Atmos. Chem. Phys., 15, 5211-5228, 10.5194/acp-15-5211-2015, 2015.

Karydis, V. A., Tsimpidi, A. P., Pozzer, A., and Lelieveld, J.: How alkaline compounds control atmospheric aerosol particle acidity, Atmos. Chem. Phys., 21, 14983-15001, 10.5194/acp-21-14983-2021, 2021.

Liu, M., Song, Y., Zhou, T., Xu, Z., Yan, C., Zheng, M., Wu, Z., Hu, M., Wu, Y., and Zhu, T.: Fine particle pH during severe haze episodes in northern China, Geophysical Research Letters, 44, 5213-5221, https://doi.org/10.1002/2017GL073210, 2017.

Myhre, G., Samset, B. H., Schulz, M., Balkanski, Y., Bauer, S., Berntsen, T. K., Bian, H., Bellouin, N., Chin, M., Diehl, T., Easter, R. C., Feichter, J., Ghan, S. J., Hauglustaine, D., Iversen, T., Kinne, S., Kirkevåg, A., Lamarque, J. F., Lin, G., Liu, X., Lund, M. T., Luo, G., Ma, X., van Noije, T., Penner, J. E., Rasch, P. J., Ruiz, A., Seland, Ø., Skeie, R. B., Stier, P., Takemura, T., Tsigaridis, K., Wang, P., Wang, Z., Xu, L., Yu, H., Yu, F., Yoon, J. H., Zhang, K., Zhang, H., and Zhou, C.: Radiative forcing of the direct aerosol effect from AeroCom Phase II simulations, Atmos. Chem. Phys., 13, 1853-1877, 10.5194/acp-13-1853-2013, 2013.

Vasilakos, P., Russell, A., Weber, R., and Nenes, A.: Understanding nitrate formation in a world with less sulfate, Atmos. Chem. Phys., 18, 12765-12775, 10.5194/acp-18-12765-2018, 2018.